# The LINC01138 drives malignancies via activating arginine methyltransferase 5 in hepatocellular carcinoma

Zhe Li[1], Jiwei Zhang[1], Xinyang Liu[3], Shengli Li[1], Qifeng Wang[1], Di Chen[1], Zhixiang Hu[1], Tao Yu[2], Jie Ding[1], Jinjun Li[2], Ming Yao[2], Jia Fan[3], Shenglin Huang [1], Qiang Gao[3], Yingjun Zhao[1] & Xianghuo He [1,4]

Recurrent chromosomal aberrations have led to the discovery of oncogenes or tumour suppressors involved in carcinogenesis. Here we characterized an oncogenic long intergenic non-coding RNA in the frequent DNA-gain regions in hepatocellular carcinoma (HCC), *LINC01138* (long intergenic non-coding RNA located on 1q21.2). The *LINC01138* locus is frequently amplified in HCC; the *LINC01138* transcript is stabilized by insulin like growth factor-2 mRNA-binding proteins 1/3 (IGF2BP1/IGF2BP3) and is associated with the malignant features and poor outcomes of HCC patients. LINC01138 acts as an oncogenic driver that promotes cell proliferation, tumorigenicity, tumour invasion and metastasis by physically interacting with arginine methyltransferase 5 (PRMT5) and enhancing its protein stability by blocking ubiquitin/proteasome-dependent degradation in HCC. The discovery of LINC01138, a promising prognostic indicator, provides insight into the molecular pathogenesis of HCC, and the LINC01138/PRMT5 axis is an ideal therapeutic target for HCC treatment.

[1] Fudan University Shanghai Cancer Center and Institutes of Biomedical Sciences, Department of Oncology, Shanghai Medical College, Fudan University, Shanghai 200032, China. [2] State Key Laboratory of Oncogenes and Related Genes, Shanghai Cancer Institute, Renji Hospital, Shanghai Jiao Tong University School of Medicine, Shanghai 200032, China. [3] Liver Cancer Institute, Zhongshan Hospital, Shanghai Medical College, Fudan University, Shanghai 200032, China. [4] Collaborative Innovation Center for Cancer Medicine, Department of Oncology, Shanghai Medical College, Fudan University, Shanghai 200032, China. These authors contributed equally: Zhe Li, Jiwei Zhang, Xinyang Liu. Correspondence and requests for materials should be addressed to Q.G. (email: gaoqiang@fudan.edu.cn) or to Y.Z. (email: zhaoyingjun@fudan.edu.cn) or to X.H. (email: xhhe@fudan.edu.cn)

Hepatocellular carcinoma (HCC) is a highly aggressive primary liver malignancy that represents the third leading cause of global cancer-related mortalities. Data statistics showed 782,500 new liver cancer cases and 745,500 deaths worldwide, with China alone accounting for ~50% of the total number of cases and deaths[1]. At the current rate, HCC will surpass breast and colorectal cancers to become the leading cause of cancer incidence in 2030[2]. Because, the molecular pathogenesis of HCC is not yet fully understood, there has been limited success in improving the disease-free survival rate of HCC patients. Thus, novel cancer-promoting genes involved in HCC must be identified and characterized to obtain a better understanding of this lethal disease and to develop clinical applications for its treatment.

Current progress in transcriptomics demonstrates that a major portion of the human transcriptome does not code for proteins. Recently, long non-coding RNAs (lncRNAs), or transcripts larger than 200 nucleotides with little or no protein-coding potential, have been discovered at an unprecedented rate. Accumulating evidence shows that lncRNAs participate in a wide range of cellular processes, including the regulation of epigenetic signatures and gene expression[3,4]. The molecular mechanisms by which lncRNAs exert their biological functions are diverse and complex, which include serving as scaffolds or guides to regulate protein–protein or protein–DNA interactions[5,6], decoys to bind to proteins[7,8] and miRNA sponges[9,10]. Emerging evidence has shown that lncRNAs regulate multiple biological processes, such as proliferation, apoptosis and cell migration, and are frequently deregulated in various human diseases, including cancer[11–13]. Notably, lncRNAs can act as oncogenes in human carcinogenesis[14,15].

In this study, we extensively analysed the differentially expressed lncRNAs in regions with recurrent chromosomal aberrations in the HCC genome and characterized a novel oncogenic long intergenic non-coding RNA (lincRNA) in HCC, *LINC01138*. LINC01138 is significantly associated with the malignant features and poor outcomes of HCC patients and exerts its biological functions through the oncogenic IGF2BP1/IGF2BP3-LINC01138-PRMT5 axis in HCC cells. Our data suggested that LINC01138 is a potentially robust biomarker and therapeutic target for HCC.

## Results

### LINC01138 is associated with poor outcomes in HCC patients.
To identify the potential oncogenic lincRNAs in HCC, we identified lincRNAs located in somatic copy number alterations (SCNAs) in HCC[16], and 53 candidate lincRNAs were selected according to the criteria (relative CNAs in >30% HCC samples, occurring in the amplification CNA area, prior to long intergenic non-coding RNA; Supplementary Data 1). With the intersection of these 53 candidate lincRNAs with relative HCC genomic gains and the 1082 lincRNAs upregulated in the TCGA cohort of 50 paired HCC tissues and adjacent non-tumour (NT) tissues, four lincRNAs were identified, namely, *LINC01138*, *PVT1*, *RP11-14N7.2* and *RP11-30J20.1* (Fig. 1a and Supplementary Fig. 1a, b). Among them, *LINC01138* (ENSG00000274020.1) showed the strongest signal intensity of CNAs for further study. Moreover, *LINC01138* showed an increased genomic copy number in a cohort of 72 HCC patients (Fig. 1b, c). *LINC01138* is located in 1q21.2 and has two major transcripts of 2075 bp and 1236 bp in length (Supplementary Fig. 2a). The results of 5′ and 3′ rapid amplification of the cDNA ends (RACE) (Supplementary Fig. 2b, c), northern blot analyses and quantitative real-time PCR (qPCR) revealed that the 2075-nt LINC01138 is the predominant and quite stable transcript in HCC cell lines and HCC tissues

(Supplementary Fig. 2d–g). In addition, the PhyloCSF codon substitution frequency analysis[17], Coding Potential Assessment Tool (CPAT), Coding Potential Calculator (CPC) and ORF finder software from the National Centre for Biotechnology Information (NCBI) indicated that LINC01138 is non-coding[18,19] (Supplementary Fig. 3a–d). In vitro transcription and translation assay showed that neither the sense nor the antisense transcript of LINC01138 could encode protein, confirming that LINC01138 is a bona fide non-coding RNA (Supplementary Fig. 3e). Moreover, LINC01138 is widely expressed in different liver cancer cell lines (Supplementary Fig. 3f) and distributes in both the cytoplasm and nucleus of SMMC-7721, SNU-449 and Huh-7 cells (Supplementary Fig. 3g). Furthermore, the relationship between LINC01138 RNA levels and the clinico-pathological features were analysed in another 120 HCC and adjacent non-cancerous tissues. The results showed that LINC01138 was apparently overexpressed which occurred in 53.3% of the HCC samples (Fig. 1d, e), and its expression was positively correlated with tumour size, alpha-fetoprotein (AFP) level, and hepatitis B surface antigen (HBsAg) level, respectively (Fig. 1f). Importantly, high LINC01138 level is remarkably associated with poor prognosis of HCC patients (Fig. 1g and Supplementary Table 1).

### LINC01138 promotes HCC cell growth and metastasis.
To further determine the oncogenic properties of LINC01138 in HCC, we used the CRISPR/dead-Cas9 system to activate the endogenous RNA level of LINC01138[20] and the CRISPR/Cas9 system to knockout the endogenous RNA level of LINC01138 (Supplementary Fig. 4a)[21]. The accumulation of endogenous LINC01138 significantly accelerated the colony formation and cell-proliferation abilities of SNU-449 and Huh-7 cells (Fig. 2a, b and Supplementary Fig. 4b), whereas knockout of endogenous LINC01138 led to a significant decrease in SMMC-7721 cells growth (Fig. 2c, d and Supplementary Fig. 4c). Additionally, LINC01138 overexpression by a lentivirus vector (pWPXL-LINC01138) promoted SNU-449 and Huh-7 cell growth and colony formation (Supplementary Fig. 4d–f); whereas, two independent small interfering RNAs (siRNAs) for LINC01138 significantly decreased SMMC-7721 cell growth and colony formation (Supplementary Fig. 4g–i).

On the other hand, CRISPR/dead-Cas9 and CRISPR/Cas9 technology confirmed the promoting function of LINC01138 on the migratory and invasive abilities of HCC cells. LINC01138 induction increased the migration and invasion of the SNU-449 and Huh-7 cells (Fig. 2e and Supplementary Fig. 5a, b), and vice-versa, LINC01138 knockout significantly decreased the migration and invasion of SMMC-7721 cells (Fig. 2f and Supplementary Fig. 5c). The results were verified by pWPXL-LINC01138 transduction or LINC01138-specific siRNAs (Supplementary Fig. 5d–g).

To further explore the growth-promoting effects of LINC01138 on HCC cells in vivo, we subcutaneously injected stable pWPXL-LINC01138 cells into nude mice. Both the volumes and weights of the tumours in the pWPXL-LINC01138 group were markedly higher than those in the control group (Fig. 2g and Supplementary Fig. 6a), demonstrating that LINC01138 promotes the tumorigenicity of the HCC cells in vivo. We also evaluated the promoting effects of LINC01138 on cell metastasis. The stable pWPXL-LINC01138 SMMC-7721 cells were transplanted into the livers of nude mice. The metastatic nodules in the livers were significantly increased in the pWPXL-LINC01138 group (Supplementary Fig. 6b). Haematoxylin-eosin staining showed that the metastatic foci derived from the pWPXL-LINC01138 cells dramatically increased in the liver, lung and intestine sections (Fig. 2h and Supplementary Fig. 6c–e). Moreover, CRISPR-

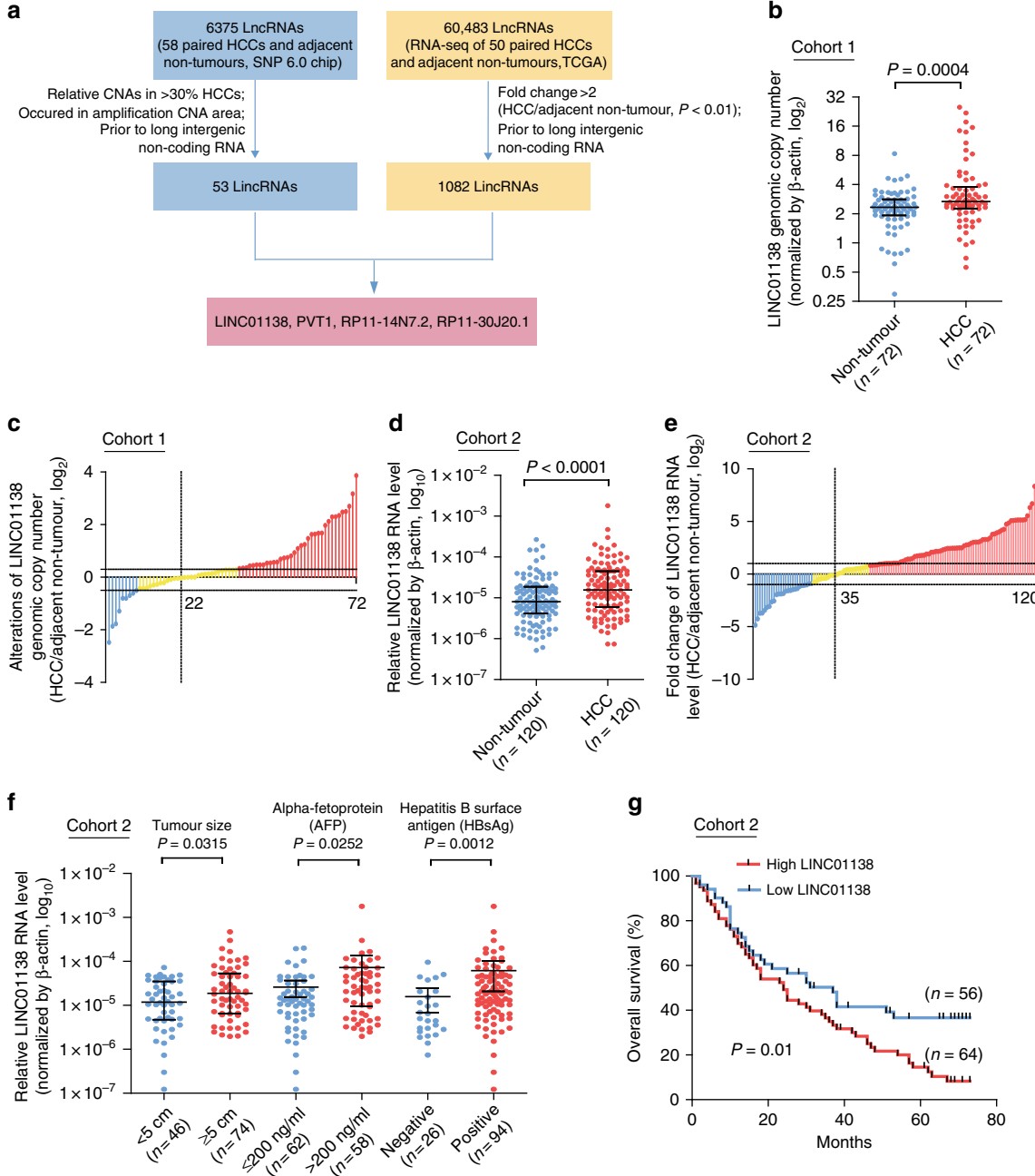

**Fig. 1** *LINC01138* is associated with clinical outcomes in patients with HCC. **a** The flow chart for selecting candidate lincRNAs in HCC; four lincRNAs were identified for further study. **b** The copy numbers of *LINC01138* were determined in 72 pairs of HCC tissues and adjacent normal tissues using qPCR. **c** Fold-change of *LINC01138* copy number variations in 72 paired tissues (deletion, blue; no-change, yellow; amplification, red.). **d** The RNA levels of *LINC01138* were quantified in 120 pairs of HCC tissues and adjacent normal tissues using qPCR. **e** Fold-changes of expression of *LINC01138* in 120 paired tissues (downexpression, blue; no-change, yellow; upexpression, red.). **f** Clinical significance of *LINC01138* in patients with HCC; high LINC01138 expression positively correlated with tumour size (≥5 cm), AFP (>200 ng/ml) and HBsAg-positive patients. **g** Kaplan–Meier analyses of the correlation between *LINC01138* RNA levels and the overall survival in 120 patients with HCC. Patients were stratified for the analysis by the median value. Values are expressed as the median with interquartile range in **b**, **d**, **f**

LINC01138-knockout cells exhibited inhibited behaviour in the in vivo assays. The metastatic nodules were significantly reduced in the CRISPR-LINC01138-KO-P1 group (Supplementary Fig. 6f). Consistent with these results, the metastatic foci derived from CRISPR-LINC01138-KO-P1 SMMC-7721 cells dramatically decreased in the liver and lung sections (Supplementary Fig. 6g, h). Taken together, these findings suggest that LINC01138 acts as an oncogenic driver in the development and progression of HCC.

**LINC01138 interacts with IGF2BP1/3 and PRMT5 in HCC cells.** To explore the molecular mechanism underlying the oncogenic activity of LINC01138 in hepatic carcinogenesis, we performed RNA pull-down assays to identify the proteins associated with LINC01138 in the HCC cells. The results from three independent LINC01138 pull-down experiments repeatedly showed specific bands at ~70 KD via mass spectrometry (Supplementary Fig. 7a). Ten potential interacting proteins were obtained based on peptide number >5 and unique peptide

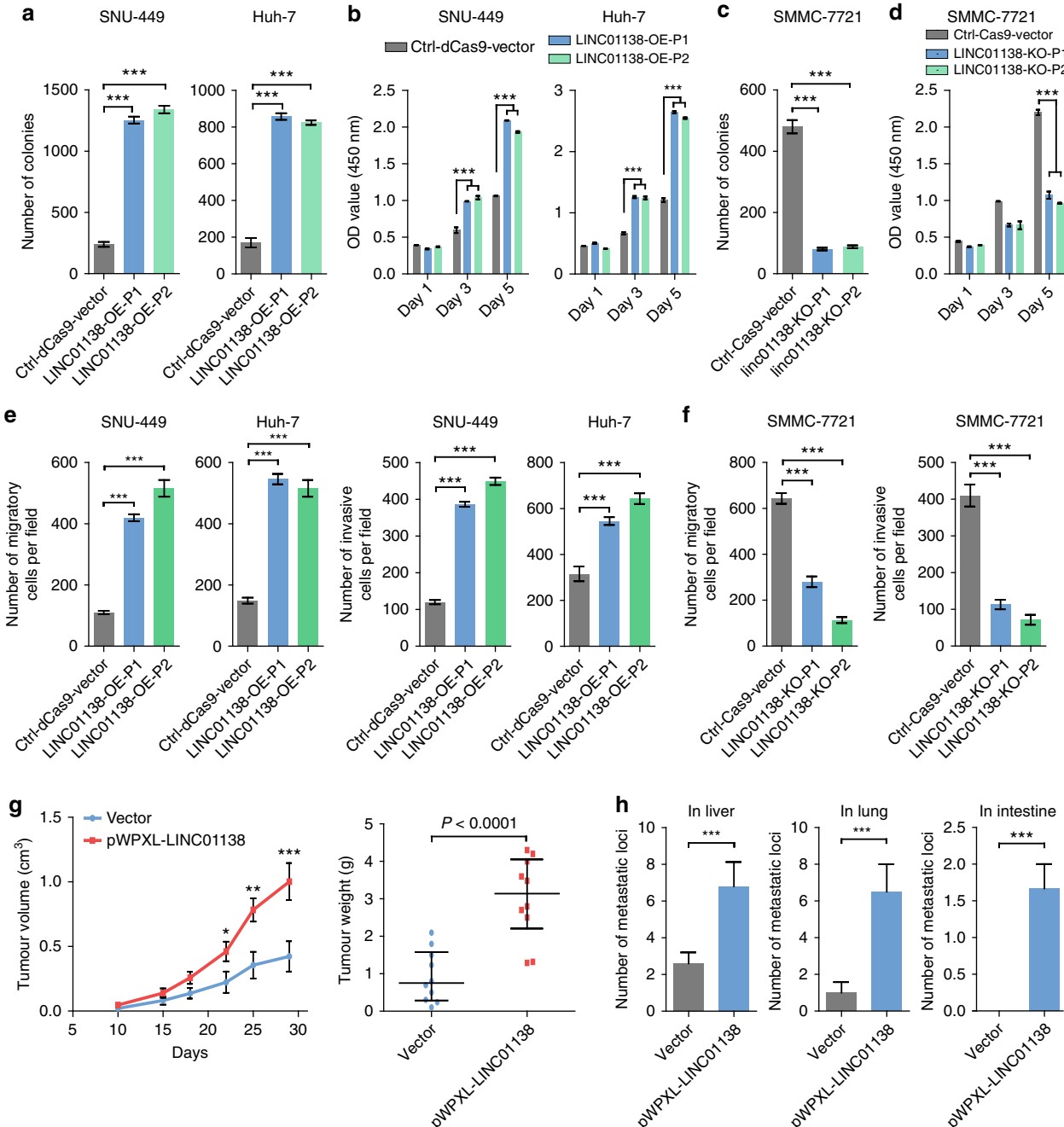

**Fig. 2** LINC01138 increases HCC cell proliferation, invasion and metastasis in vitro and in vivo. **a**, **b** Colony formation assays (**a**) and CCK-8 assays (**b**) in the SNU-449 and Huh-7 cells with LINC01138 activation by the CRISPR/dead-Cas9 technology. **c**, **d** Colony formation assays (**c**) and CCK-8 assays (**d**) in the SMMC-7721 cells with LINC01138 knockout by the CRISPR/Cas9 technology. **e** Transwell migration and invasion assays in the SNU-449 and Huh-7 cells with LINC01138 activation by the CRISPR/dead-Cas9 technology. **f** Transwell migration and invasion assays in the SMMC-7721 cells with LINC01138 knockout by the CRISPR/Cas9 technology. **g** Tumour volumes and tumour weights were measured in the pWPXL-LINC01138 and negative control groups in the xenograft mouse models. **h** Statistics analysis of the metastatic foci in the liver, lung and intestine obtained from nude mice at 9 weeks after injection with the stable pWPXL-LINC01138 SMMC-7721 cells detected by haematoxylin-eosin staining. The stably cells established via the CRISPR/Cas9 knockout and CRISPR/dead-Cas9 activation technology was nominated as KO and OE, respectively. Two paired sgRNAs were identified to be effective in both knockout system and activation system, nominated as P1 and P2, respectively. Values are expressed as the mean ± SEM, $n = 3$ in **b–f**, $n = 10$ in **g**, **h**. ***$P < 0.001$

number >5 in the three independent experiments and were absent in the corresponding antisense groups (Supplementary Table 2). After confirming in three independent experiments, we observed that sense but not antisense LINC01138, was specifically associated with IGF2BP1, IGF2BP3 and PRMT5 (Fig. 3a). Moreover, RIP assays showed that the antibodies of

IGF2BP1, IGF2BP3 or PRMT5 could significantly enrich LINC01138 (Fig. 3b); whereas, the GAPDH antibody and IgG control could not. As expected, there was no obvious enrichment of PVT1 in each group (the right of Fig. 3b). Furthermore, a series of deletion was constructed based on the secondary structure of LINC01138 (http://www.lncipedia.org/,

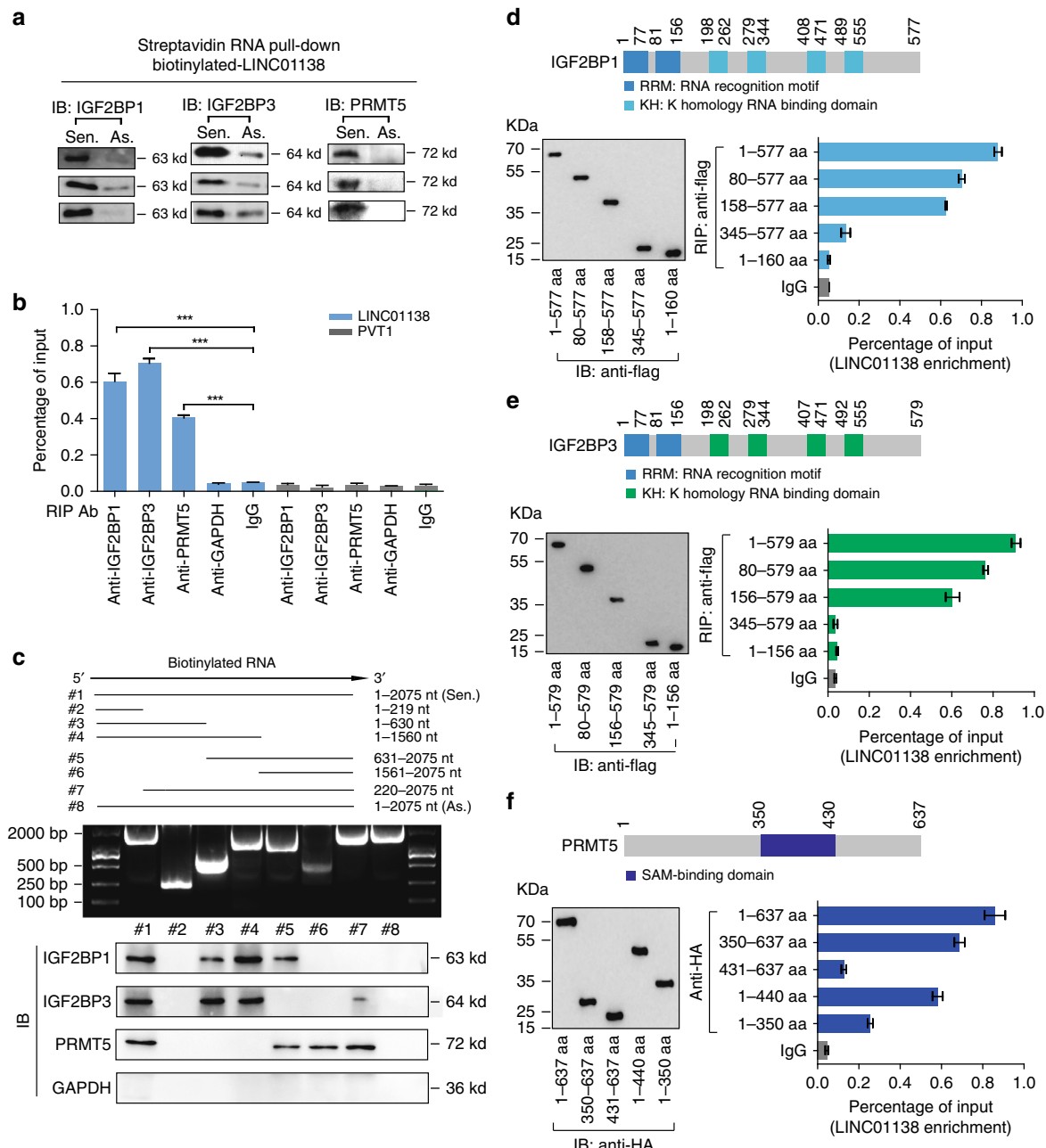

**Fig. 3** LINC01138 physically interacts with IGF2BP1, IGF2BP3 and PRMT5 in HCC cells. **a** Immunoblotting for the specific associations of IGF2BP1, IGF2BP3 or PRMT5 with biotinylated-LINC01138 from three independent streptavidin RNA pull-down assays. **b** RIP assays were performed using the indicated antibodies. Real-time PCR was used to detect LINC01138 enrichment, using GAPDH antibody as the antibody control and 2036-nt PVT1 as the LincRNA control. **c** Immunoblotting of IGF2BP1, IGF2BP3 or PRMT5 in pull-down samples by full-length biotinylated-LINC01138 (#1) or truncated biotinylated-LINC01138 RNA motifs (#2: 1–219 nt; #3: 1–630 nt; #4: 1–1560 nt; #5: 630–2075 nt; #6: 1560–2075 nt; #7: 220–2075 nt), with GAPDH as the negative control. **d–f** Deletion mapping for the domains of IGF2BP1 (**d**), IGF2BP3 (**e**) or PRMT5 (**f**) that bind to LINC01138. RIP analysis for LINC01138 enrichment in cells transiently transfected with plasmids containing the indicated FLAG-tagged or HA-tagged full-length or truncated constructs. Values are expressed as the mean ± SEM, *n* = 3 in **b**, **d–f**

Supplementary Fig. 7b). The 220–1560-nt fragment of LINC01138 mediates the interaction with IGF2BP1 or IGF2BP3, and the fragments, which could interact with both IGF2BP1 and IGF2BP3, share the 220–630-nt sequence of Linc01138, while the 1561–2075-nt fragment of LINC01138 is required and sufficient for the association with PRMT5 (Fig. 3c). Next, RIP assays for FLAG-tagged full-length and truncated IGF2BP1 showed that the deletion of some K homology RNA-binding domains (KH, 158–345 aa) of IGF2BP1 significantly abolished the association between this

protein and LINC01138 (Fig. 3d). Additionally, the deletion of some KH domains (156–345 aa) of IGF2BP3 exhibit similar effects (Fig. 3e), indicating that LINC01138 binds to these regions. The S-adenosylmethionine (SAM)-binding domain (350–430 aa) of PRMT5 physically associates with LINC01138 in the HCC cells (Fig. 3f), and the LINC01138-binding abilities of the mutants without this domain were definitely dampened. Together, these results indicated that LINC01138 specifically binds with IGF2BP1, IGF2BP3 and PRMT5 in HCC cells.

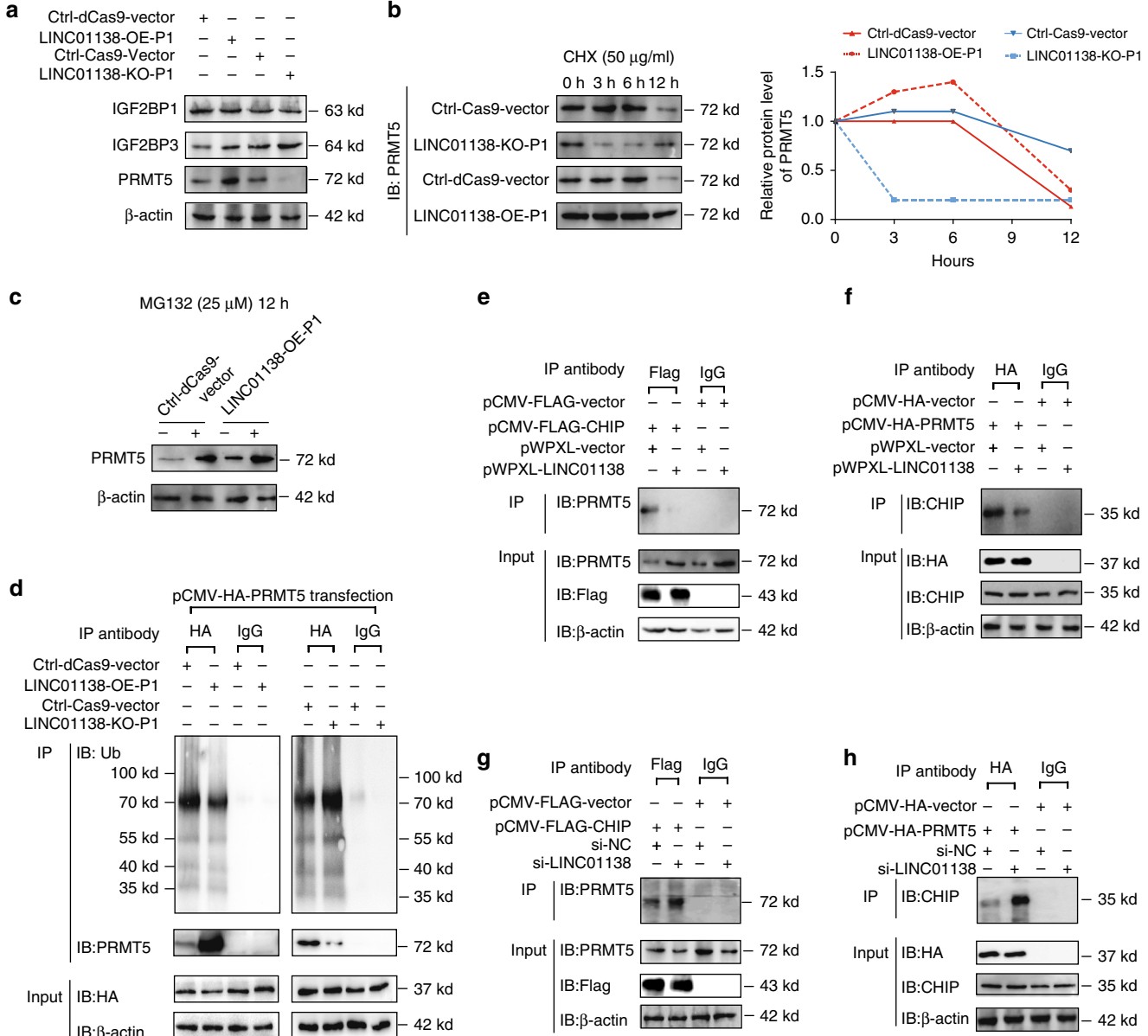

**Fig. 4** LINC01138 blocks ubiquitin/proteasome-dependent PRMT5 degradation. **a** Immunoblotting for the protein levels of IGF2BP1, IGF2BP3 and PRMT5 after LINC01138 activation or knockout. β-actin served as the internal control. **b** LINC01138-KO cells, LINC01138-OE cells, or the control cells were treated with cycloheximide (CHX, 50 μg/ml) for the indicated times. Left, immunoblotting for the PRMT5 levels in whole-cell extracts; right, the densitometry analysis of the PRMT5 protein levels; the relative fold of the level at 0 h. **c** LINC01138-OE or vector cells were treated with MG132 (25 μM) for 12 h. Immunoblotting for PRMT5 levels in the indicated cells. **d** LINC01138-KO or LINC01138-OE cells were transfected with pCMV-HA-PRMT5 plasmids for 48 h. The cell lysates were immunoprecipitated (IP) with either control IgG or HA antibody and immunoblotted with the ubiquitin-specific antibody. HA-tagged PRMT5 and β-actin served as the loading control. **e–h** Immunoprecipitation to detect the association between PRMT5 and CHIP after LINC01138 overexpression or knockdown. pCMV-HA-PRMT5 plasmid and pCMV-Flag-CHIP plasmid were co-transfected into pWPXL-LINC01138 or pWPXL-Vector SMMC-7721 cells (**e**, **f**) and si-NC or si-LINC01138 SMMC-7721 cells (**g**, **h**) for 48 h. FLAG-tagged CHIP, HA-tagged PRMT5 and β-actin served as the loading control

**LINC01138 blocks PRMT5 ubiquitination and degradation.** Given that LINC01138 interacts with IGF2BP1, IGF2BP3 and PRMT5 in HCC cells, we characterized the molecular consequences of these associations. Interestingly, LINC01138 had no effect on the protein and mRNA levels of IGF2BP1 and IGF2BP3 (Fig. 4a and Supplementary Fig. 8a, b). Although LINC01138 had no significant effect on the mRNA level of PRMT5, and vice-versa (Supplementary Fig. 8c, d), the protein levels of PRMT5 were dramatically reduced when silencing LINC01138 and were increased when overexpressing LINC01138 (Fig. 4a and

Supplementary Fig. 8e). Moreover, following treatment with a protein-synthesis inhibitor cycloheximide (CHX), LINC01138 knockout decreased the half-life of the PRMT5 protein, whereas LINC01138 activation increased the half-life of the PRMT5 protein in the HCC cells (Fig. 4b and Supplementary Fig. 8f), with β-actin as the relative control (Supplementary Fig. 8g). Following treatment with a proteasome inhibitor MG132, the accumulation of endogenous PRMT5 in cells overexpressing LINC01138 was greater (Fig. 4c and Supplementary Fig. 8h), indicating that LINC01138 might inhibit the proteasome-dependent degradation

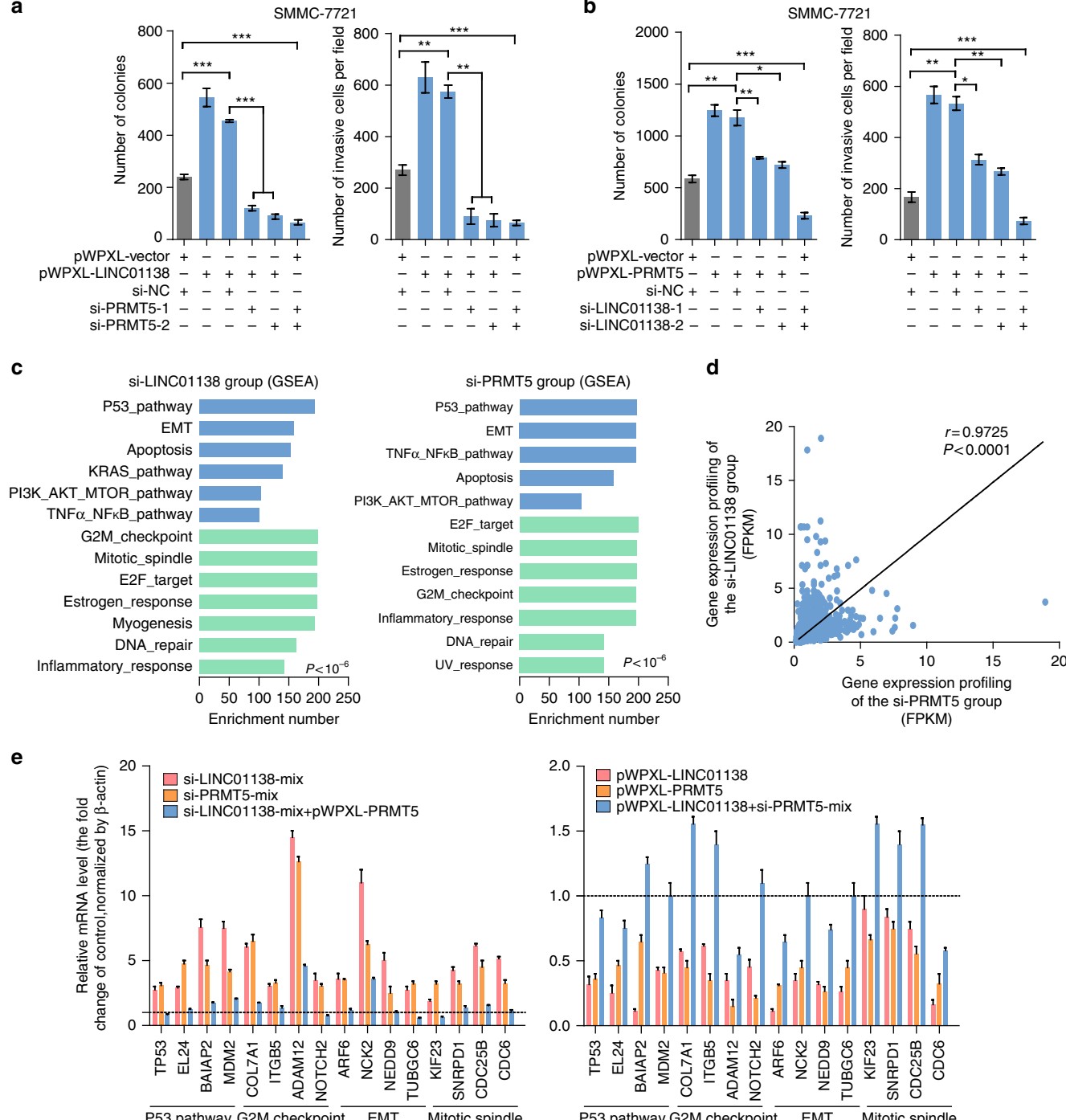

**Fig. 5** PRMT5 mediates LINC01138-driven HCC cell growth, migration and invasion. **a** Rescue assays of colony formation assay were performed after silencing PRMT5 in pWPXL-LINC01138 cells. **b** Rescue assays of transwell invasion assay were performed after silencing PRMT5 in pWPXL-LINC01138 cells. **c** GSEA enrichment focused on a set of signalling pathways and biological processes after LINC01138 silencing and PRMT5 silencing, summarized based on the enrichment score (signalling pathway, blue; biological process, green). **d** Correlation analysis of the gene expression profiling between the si-LINC01138 group and si-PRMT5 group. **e** The mRNA levels of tumorigenesis-related genes in the high-scored four signalling pathways in cells with indicated treatments. Values are expressed as the mean ± SEM, $n = 3$ in **a**, **b**, **d**, **e**. **$P < 0.01$ and ***$P < 0.001$

of PRMT5 in HCC cells. Furthermore, the ubiquitination levels of PRMT5 significantly decreased in the LINC01138-OE-P1 cells; whereas, the ubiquitination levels of PRMT5 increased in the LINC01138-KO-P1 cells (Fig. 4d). A recent report, which identified CHIP as an E3 ligase for PRMT5 ubiquitination and degradation[22], prompted us to detect the function of LINC01138 in the interaction between CHIP E3 ligase and PRMT5 in the

HCC cells. Indeed, LINC01138 overexpression significantly inhibited the association between CHIP and PRMT5 (Fig. 4e, f), whereas LINC01138 knockdown notably increased this association in the HCC cells (Fig. 4g, h). Collectively, these results indicated that LINC01138 can increase the stability of PRMT5 through blocking its ubiquitin/proteasome-dependent degradation.

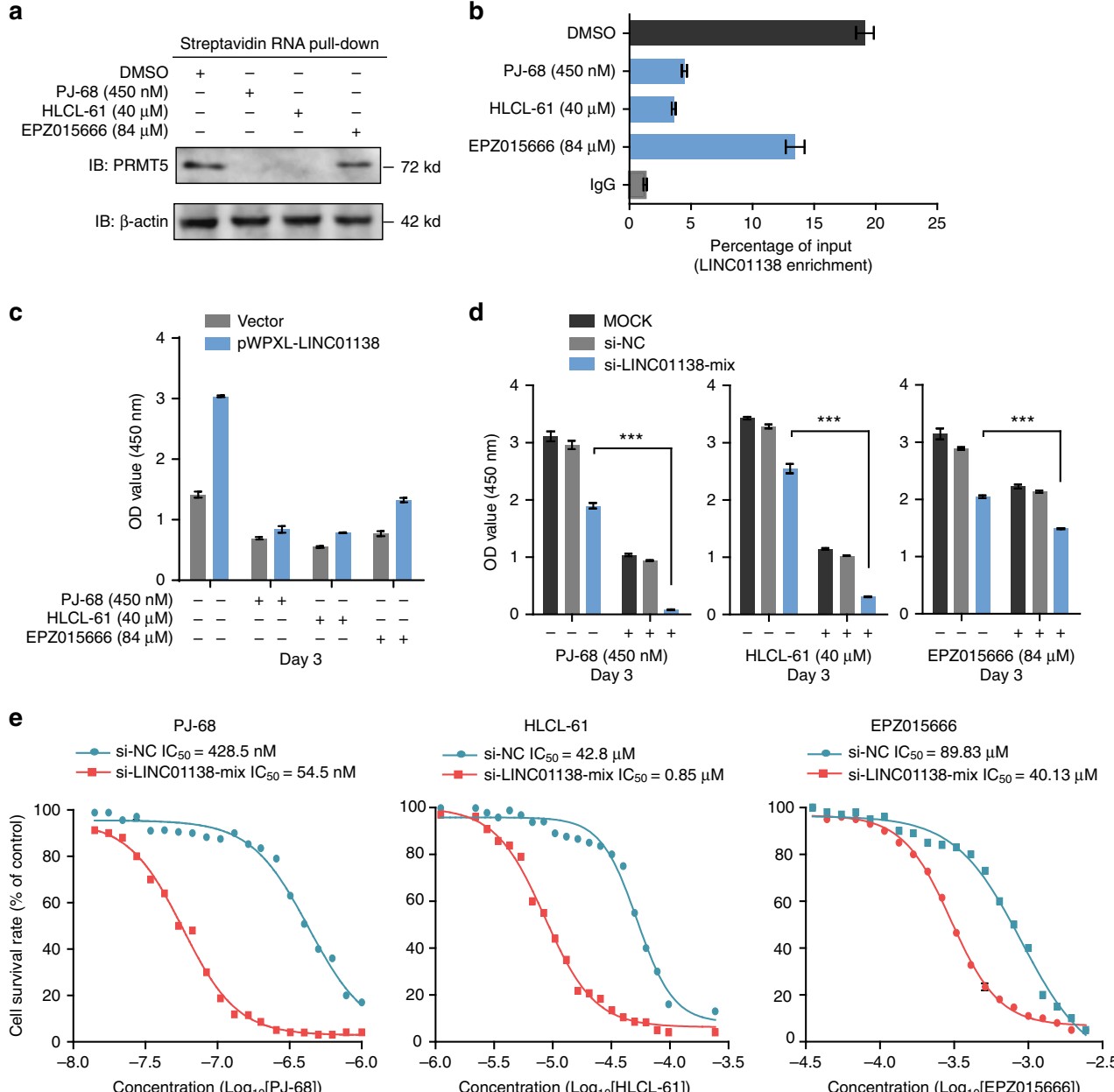

**Fig. 6** The association between LINC01138 and PRMT5 is a candidate therapeutic target for HCC. **a** RNA pull-down assays for the specific association of PRMT5 and LINC01138, in the cells exposed to PRMT5 inhibitors at the indicated concentration for 48 h. **b** SMMC-7721 cells were transfected with plasmids containing HA-tagged PRMT5, and RIP assays were performed using HA antibodies in the cells treated with PRMT5 inhibitors at the indicated concentration for 48 h. **c** CCK-8 assays for pWPXL-LINC01138 or vector cells, exposed to PRMT5 inhibitors at the indicated concentration for 3 days. **d** CCK-8 assays in cells transfected with si-NC or si-LINC01138 mixture, exposed to PRMT5 inhibitors at the indicated concentration for 3 days. **e** $IC_{50}$ were analysed in cells transfected with si-NC or si-LINC01138 mixture, exposed to PRMT5 inhibitors for 3 days. Values are expressed as the mean ± SEM, $n = 3$ **b**–**e**. $^{**}P < 0.01$ and $^{***}P < 0.001$

**PRMT5 is a functional downstream mediator for LINC01138.** Considering the effect of LINC01138 on PRMT5 protein stability, we hypothesize that LINC01138 may exert its biological effects through stabilizing PRMT5. Specific siRNAs for PRMT5 remarkably reduced the colony formation, proliferation and invasion abilities of the SMMC-7721 and SK-Hep-1 cells (Supplementary Fig. 9a–f). In addition, PRMT5 overexpression significantly increased the invasion abilities of these cells (Supplementary Fig. 9g–i), confirming that PRMT5 is a tumour-promoting gene in HCC. Subsequently, PRMT5 siRNAs remarkably impaired the colony formation and invasion abilities induced by LINC01138 overexpression (Fig. 5a and

Supplementary Fig. 10a, b), and PRMT5 overexpression via lentivirus could restore the colony formation and invasion abilities which had been reduced by LINC01138 siRNAs (Fig. 5b and Supplementary Fig. 10c, d), demonstrating the contribution of PRMT5 in the LINC01138-induced effects. In addition, unbiased transcriptome profiling was performed using RNA-sequencing in SMMC-7721 cells transfected with si-LINC01138 or si-PRMT5 to investigate the related signalling pathways in HCC cells. The depletion of LINC01138 affected the expression levels of 457 genes, of which 239 genes were upregulated and 218 genes were downregulated (fold-change >2, Supplementary Fig. 11a and Supplementary Data 2). RNA-seq data set of si-PRMT5 showed

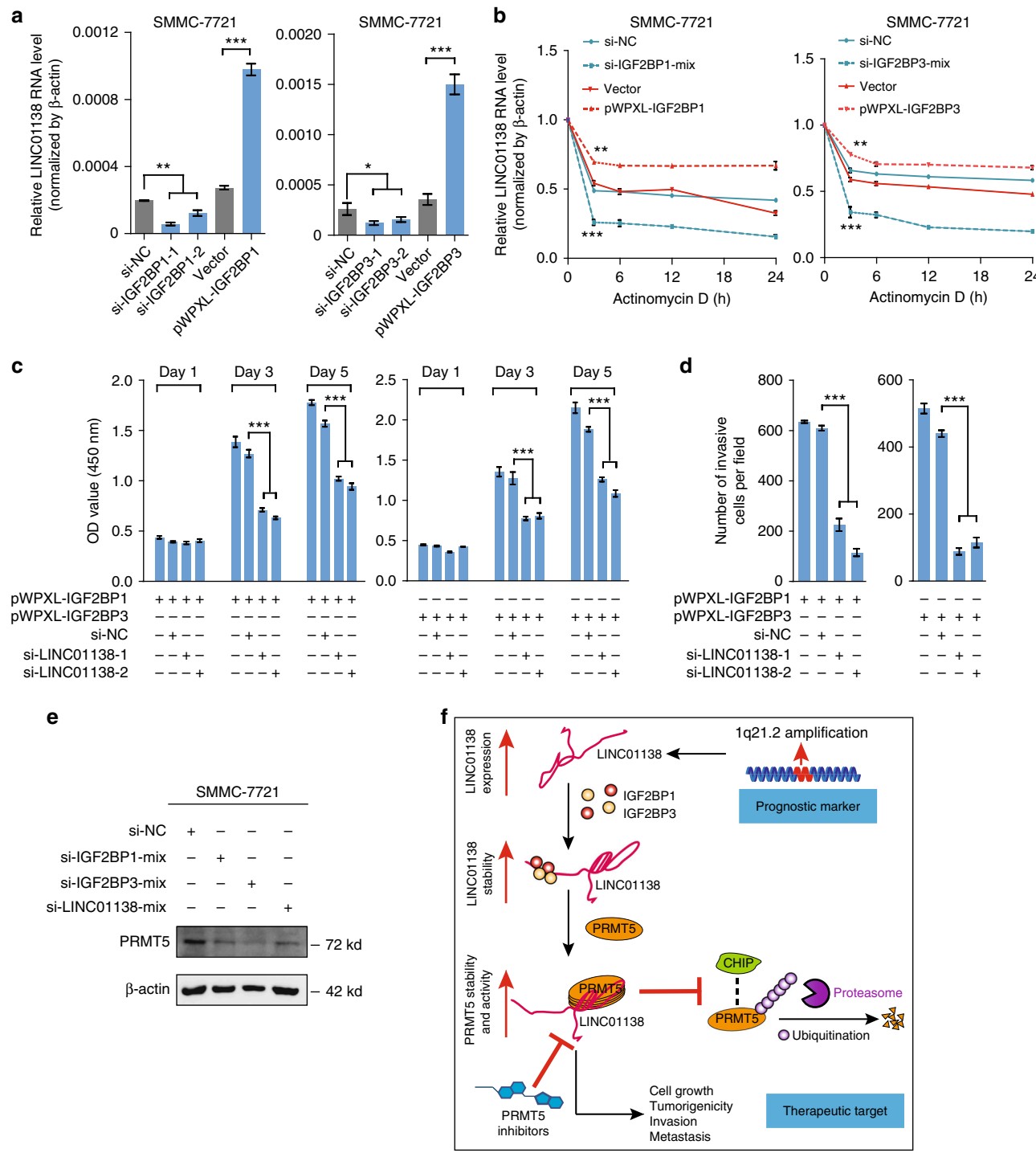

**Fig. 7** The oncogenic IGF2BP1/IGF2BP3-LINC01138-PRMT5 axis in HCC cells. **a** Relative RNA levels of LINC01138 in SMMC-7721 cells with IGF2BP1/IGF2BP3 knockdown or overexpression, using qPCR. **b** The half-life of LINC01138 after treatment with 2.5 μM actinomycin D for indicated times, with IGF2BP1/IGF2BP3 knockdown or overexpression in SMMC-7721 cells. **c** CCK-8 rescue assays were performed after LINC01138 knockdown in pWPXL-IGF2BP1 or pWPXL-IGF2BP3 cells. **d** Transwell invasion rescue assays were performed after LINC01138 knockdown in pWPXL-IGF2BP1 or pWPXL-IGF2BP3 cells. **e** Immunoblotting to detect the protein levels of PRMT5 after knockdown of IGF2BP1/IGF2BP3 or LINC01138 in SMMC-7721 cells. **f** Integrated model depicting lncRNA LINC01138 as an oncogene in liver cancer. The working model shows that LINC01138 loci is amplified, and its transcript is stabilized by IGF2BP1 or IGF2BP3 in HCC; LINC01138 exerts its oncogenic activity through interacting with and stabilizing PRMT5, which can be disrupted by small molecule inhibitors in HCC. Values are expressed as the mean ± SEM, $n = 3$ in **a-d**. ***$P < 0.001$

that the expression levels of 510 genes were changed; among these genes, 237 were upregulated and 273 were downregulated (fold-change >2, Supplementary Fig. 11b and Supplementary Data 3). In contrast with these two RNA sequencing data sets, the Gene set enrichment analysis (GSEA) analysis showed that both LINC01138 and PRMT5 share highly similar downstream signalling pathways involved in cell proliferation, the cell cycle and metastasis in HCC cells (Fig. 5c). The GSEA additionally revealed that the coincident top four gene sets included the P53 pathway, epithelial mesenchymal transition (EMT), G2M checkpoint and

mitotic spindle (Supplementary Fig. 11c, d). Importantly, there is a strong positive correlation between the gene expression profiling of the si-LINC01138 group and that of the si-PRMT5 group ($r = 0.9725$, $P < 0.0001$; Fig. 5d), with TSLNC8 as the negative lincRNA control ($r = 0.0678$, $P = 0.6930$; Supplementary Fig. 11e)[23]. Moreover, we verified the top-scoring genes altered in these two databases and confirmed that LINC01138 and PRMT5 dramatically affected the same genes that are highly associated with tumorigenesis (Fig. 5e). Furthermore, PRMT5 restoration remarkably reversed the effects of si-LINC01138 on the expression of these detected genes (the left side of Fig. 5e). On the other hand, PRMT5 knockdown significantly rescued the expression levels of the genes inhibited by LINC01138 (the right side of Fig. 5e), supporting that LINC01138 regulates the expression of downstream genes through modulating PRMT5. Taken together, these findings demonstrated that LINC01138 is an oncogenic driver through activating PRMT5 in HCC cells.

**The LINC01138/PRMT5 axis is a candidate therapeutic target.** Importantly, PRMT5 is a promising therapeutic target for cancer treatment, and several selective and specific small-molecule inhibitors for PRMT5, i.e., PJ-68 and HLCL-61, can inhibit PRMT5 activity through overlapping the enzymatic active site of the "double-E" loop or pyridine ring in the SAM-domain[24,25]. We evaluated whether these inhibitors could be used to targeting the LINC01138-PRMT5 axis in HCC cells. EPZ015666 is a SAM-uncompetitive, peptide-competitive inhibitor of the PRMT5/MEP50 complex formation to inhibit PRMT5 activity, used as a negative control for targeting the SAM domain. RNA pull-down assays showed that LINC01138 failed to interact with PRMT5 after treatment with PJ-68 or HLCL-61(Fig. 6a), whereas EPZ015666 showed no significant effects on this association in the SMMC-7721 cells. RIP assays confirmed the interference of PJ-68 or HLCL-61 treatments in the interaction between LINC01138 and PRMT5 in the SMMC-7721 cells (Fig. 6b). All three inhibitors could significantly inhibit the cell proliferation of the LINC01138-overexpressing HCC cells (Fig. 6c), suggesting that PRMT5 is an additional promising therapeutic target in HCC with high LINC01138 levels. The $IC_{50}$ values (concentrations of inhibitor that leads to a 50% reduction in viability) of PJ-68 ($IC_{50}$ = 601.7 nM), HLCL-61 ($IC_{50}$ = 50.38 μM) and EPZ015666 ($IC_{50}$ = 97.62 μM) were slightly higher in the LINC01138-overexpressing SMMC-7721 cells than those in the control cells (Supplementary Fig. 12a–c). Moreover, with silencing of endogenous LINC01138, the treatments with PJ-68 or HLCL-61 led to a sharp reduction in HCC cell proliferation (Fig. 6d), whereas a slight decrease was observed after EPZ015666 administration with LINC01138 siRNAs. Notably, the $IC_{50}$ of PJ-68 ($IC_{50}$ = 54.5 nM) and HLCL-61 ($IC_{50}$ = 0.85 μM) in LINC01138-silenced cells were dramatically decreased, ~12.7% and 2% compared with the values in the control cells, respectively, while the $IC_{50}$ of EPZ015666 ($IC_{50}$ = 40.13 μM) only decreased to ~50% (Fig. 6e). These findings revealed that the LINC01138-PRMT5 association could be a potential therapeutic target in HCC.

**IGF2BP1 or IGF2BP3 enhances LINC01138 stability in HCC cells.** The results of the present study demonstrated that LINC01138 did not affect the expression of IGF2BP1 and IGF2BP3 in the HCC cells, prompting us to evaluate whether LINC01138 could be regulated by IGF2BP1/IGF2BP3 in the HCC cells. Silencing IGF2BP1 or IGF2BP3 significantly decreased the RNA levels of LINC01138, whereas overexpression of IGF2BP1 or IGF2BP3 enhanced the LINC01138 RNA levels in the SMMC-7721 cells (Fig. 7a and Supplementary Fig. 13a, b). Moreover, actinomycin D, which effectively inhibits the de novo synthesis of

RNA, was used to explore the stability of LINC01138. Over-expression of IGF2BP1 or IGF2BP3 could increase the half-life and steady-state level of LINC01138; whereas, the depletion of IGF2BP1 or IGF2BP3 resulted in a decreased half-life and RNA level of LINC01138 (Fig. 7b), revealing that IGF2BP1 and IGF2BP3 specifically regulate the stability of LINC01138 in the HCC cells. Furthermore, knockdown of IGF2BP1 or IGF2BP3 reduced the proliferation, colony formation and invasion abilities of the SMMC-7721 cells (Supplementary Fig. 13c–h). Deleting LINC01138 in the pWPXL-IGF2BP1 cells or pWPXL-IGF2BP3 cells significantly inhibited their proliferation and invasion abilities (Fig.7c, d and Supplementary Fig. 13i, j), indicating that LINC01138 is an important downstream effector of IGF2BP1 and IGF2BP3 in HCC cells. Although, neither IGF2BP1 nor IGF2BP3 interacts with PRMT5 (Supplementary Fig. 13k), IGF2BP1 and IGF2BP3 regulated the protein level of PRMT5 by stabilizing LINC01138 in the HCC cells (Fig. 7e), suggesting the existence of the IGF2BP1/IGF2BP3-LINC01138-PRMT5 flow in the HCC cells. All these data support a model whereby LINC01138 acts as a crucial effector and regulator in the oncogenic IGF2BP1/IGF2BP3-LINC01138-PRMT5 axis to maintain high steady-state levels of HCC cell proliferation and metastasis (Fig. 7f).

**Discussion**
Recurrent chromosomal copy number alterations are often observed in HCC and typically lead to the activation of oncogenes or inactivation of tumour suppressors in hepatic carcinogenesis. In this study, we identified four differentially expressed lincRNAs in the regions of frequent DNA copy number gains in HCC. Among them, *PVT1*, a well-known long non-coding RNA, has been reported in multiple studies[15,26]. Among the remaining three novel lincRNAs, *LINC01138*, which is located at chromosome 1q21.2, is most frequently amplified in HCC. *LINC01138* displays a remarkable trend of increased expression in HCC tissues and shows positive correlations with the clinico-pathological features of HCC. Importantly, higher LINC01138 levels predicted lower overall survival rates in patients with HCC, supporting that LINC01138 may be a promising prognostic biomarker for HCC. LINC01138 showed strong oncogenic activity by promoting HCC cell proliferation, tumorigenicity, tumour invasion and metastasis in vitro and in vivo. A recent report validated LINC01138 as a prognostic indicator in prostatic cancer[27]. These findings indicate that *LINC01138* may act as a broad-spectrum tumour-promoting gene, in addition to promoting HCC.

LncRNAs typically exert their biological functions through physical interactions with regulatory proteins, miRNAs or other cellular factors[28]. In this study, we identified IGF2BP1, IGF2BP3 and PRMT5 as the bona fide interacting partners of LINC01138 and demonstrated the oncogenic function of the IGF2BP1/IGF2BP3- LINC01138-PRMT5 axis in HCC cells. IGF2BP1 and IGF2BP3 bind to the left arm of LINC01138 (220–1560-nt region in the 5′ terminus); whereas, the right arm of LINC01138 (1561–2075-nt in the 3′ terminus) accounts for its association with PRMT5; therefore, it was not surprising that these three proteins do not bind to each other and execute a distinguished role in the oncogenic axis. IGF2BP1 and IGF2BP3 have been reported in multiple types of cancers as stability regulators of multiple mRNAs. Specifically, IGF2BP1 stabilizes MYC, MDR1 and PTEN mRNAs[29,30], and IGF2BP3 promotes the expression of specific target genes (including IGF2, HMGA2, CCND1, MMP9 and CD44) by either preventing mRNA decay or stimulating mRNA translation[31,32]. There is evidence that IGF2BP1 and IGF2BP3 were overexpressed in HCC and promoted HCC cell growth[33,34]. We showed that IGF2BP1 and IGF2BP3 have a strong impact on Linc01138 RNA stability, and interacting with

and stabilizing LINC01138 might contribute to the promoting effects of these two proteins on HCC cell proliferation, colony formation and migration. Notably, our study enriched the understanding of the regulation of LINC01138 expression, occurring at the genomic level (copy number amplification), transcriptional level (activated by AR)[27], and post-transcriptional level (stabilized by IGF2BP1 and IGF2BP3) in HCC cells.

Additionally, we identified PRMT5 as the downstream effector of LINC01138. PRMT5 is a member of the arginine methyltransferase (PRMTs) protein family and mediates the methylation of protein substrates containing arginine residues[35]. PRMT5 directly methylates the R30 residues of the p65 subunit and regulates the expression of NF-κB-dependent genes[36]. In particular, PRMT5 plays multiple roles at the transcriptional or post-transcriptional levels through the arginine methylation of different downstream substances under certain conditions[37,38]. As an oncogene, PRMT5 serves critical functions in several cellular responses, including the development, progression, and aggressiveness of cancer, and has been studied in multiple tumour types, including leukaemia, lymphoma, lung cancer, liver cancer, colorectal cancer and breast cancer[39–43]. In this study, we confirmed the oncogenic effects of PRMT5 on hepatic carcinogenesis and provided a new regulatory mechanism for PRMT5 via LINC01138. LINC01138 and PRMT5 share highly similar patterns of downstream signalling pathways in cell proliferation, cell cycle and metastasis, supporting that PRMT5 functions as the downstream effector of LINC01138. Importantly, several small selective pharmaceutical inhibitors for PRMT5 were developed for cancer treatment[24,25,44]. The inhibitors PJ-68 and HLCL-61 could specifically inhibit the activity of PRMT5 in leukaemia, through binding with the enzymatic active site in the SAM-domain[24,25], which is the domain responsible for the interaction of PRMT5 with LINC01138. These specific inhibitors could abolish the association between LINC01138 and PRMT5 by competitively binding to the SAM domain of PRMT5 and result in an additive inhibitory effect on HCC cell proliferation when combined with LINC01138 knockdown. Knockdown of endogenous LINC01138 could greatly sensitize HCC cells to these pharmaceutical inhibitors; whereas, the inhibitor EPZ015666 could not induce this improved therapeutic effect without its ability to bind to the PRMT5 SAM-domain. Our results strongly support that an active site-binding feature is crucial for the specific inhibitor to abolish the association between LINC01138 and PRMT5 competitively, such as PJ-68 and HLCL-61, thus leading to an additive therapeutic effect that involves targeting the LINC01138-PRMT5 association. Our findings suggest that the newly identified IGF2BP1/IGF2BP3-LINC01138-PRMT5 axis is a potential therapeutic target for HCC, and the combination of specific small molecule inhibitors for PRMT5 and LINC01138 can be used to treat HCC patients with high expression of LINC01138.

In conclusion, our findings showed that the lincRNA *LINC01138* on 1q21.2 is amplified, overexpressed, and stabilized by IGF2BP1 and IGF2BP3 in HCC cells. *LINC01138* acts as an oncogene and significantly promotes HCC cell proliferation, tumorigenicity, tumour invasion and metastasis in vitro and in vivo. LINC01138 physically interacts with PRMT5 and exerts its oncogenic activity by stabilizing PRMT5 in HCC cells; specific PRMT5 inhibitors can disturb the association between LINC01138 and PRMT5, thus suppressing HCC cell growth. The discovery of *LINC01138*, a promising prognostic indicator, provides insight into hepatic carcinogenesis and may facilitate the development of precise approaches for cancer screening and treatment.

## Methods

**Patients and ethical statement.** Fifty paired HCC and corresponding adjacent non-tumour (NT) liver tissues from TCGA database (https://cancergenome.nih.gov/); Cohort 1:72 paired patients HCC and corresponding adjacent NT liver tissues were obtained from the surgical specimen archives of the First Affiliated Hospital of Zhejiang University; Cohort 2:120 paired patients HCC and corresponding adjacent NT liver tissues were obtained from the surgical specimen archives of the Zhongshan Hospital, Shanghai, China. A summary of the clinical information for the 120 patients is available online in Supplementary Table 1. The tissue specimens were snap-frozen in liquid nitrogen and stored at −80 °C for RNA or protein extraction. The use of human clinical specimens in the present study was approved by the Institutional Review Board of the Shanghai Medical College of Fudan University. The patients were informed, and they signed consent forms acknowledging the use of their resected tissues for research purposes. A pathologist graded and analysed the clinico-pathological features of HCC patients, including venous invasion, tumour encapsulation, tumour microsatellite formation, hepatitis B surface antigen, direct liver invasion, cellular differentiation, tumour size and pathological tumour-node-metastasis (pTNM) tumour stage.

**Cell lines and reagents.** HEK293T and SNU-449 cells were obtained from the American Type Culture Collection (ATCC, Manassas, Virginia, USA). Huh-7 cells were obtained from the Japanese Collection of Research Bioresources (JCRB, Tokyo, Japan). SMMC-7721 cell line was purchased from the Shanghai Cell Bank Type Culture Collection Committee (CBTCCC, Shanghai, China). All cells were cultured at 37 °C in a humidified incubator with 5% $CO_2$ in Dulbecco's modified Eagle's medium (DMEM) (Invitrogen, Carlsbad, CA, USA) or Roswell Park Memorial Institute (RPMI) 1640 medium (Invitrogen) supplemented with 10% foetal bovine serum (FBS) (HyClone, Logan, UT, USA), 100 U/ml penicillin and 100 μg/ml streptomycin (Invitrogen), and 8 mg/l antibiotic tylosin tartrate against mycoplasma (Sigma-Aldrich, St. Louis, Missouri, USA). Cell lines were authenticated by short tandem repeats (STR) profiling and confirmed to be mycoplasma-free. All the cell lines were used within 20 passages and subjected to routine cell line quality examinations (e.g., morphology, mycoplasma), and thawed fresh every 2 months.

Cycloheximide (CHX) and the proteasome inhibitor MG132 were purchased from Sigma-Aldrich. Small molecule inhibitors HLCL-61 was purchased from Selleckchem (Selleckchem, Shanghai, China).

**Genomic DNA copy number alteration analysis.** A SNP array data (Affymetrix Genome-Wide Human SNP Array 6.0) of 58 paired HCC and non-tumour tissues was used to predict the gain or loss for each gene, inferred for a relative copy number[16]. In brief, the copy number was calculated based on the hybridization signal intensity, compared to the signal in adjacent non-tumour tissue from the same patient. Significant CNAs were detected using a hidden Markov model (HMM) algorithm in the standard Partek workflow for paired samples, with the criteria as follows: the minimum physical length of the putative CNA was no <100 kb; a gain or loss was inferred for a relative copy number of >2.6 or <1.3, respectively; the CNA was present in at least three tumour samples; and the overlapping common regions among multiple tumours were calculated. Gene annotation and overlap were performed using the University of California, Santa Cruz (UCSC) hg38 to assemble the lncRNAs in the regions with significant genomic gains. Real-time qPCR was performed to determine the relative copy number of the target genes in paired HCC tissues and the corresponding adjacent non-tumour tissues, using SYBR Green (Takara, Japan) in the 7900HT Fast Real-Time PCR System (Applied Biosystems, USA). The number of copies of the target genes in each test sample is determined by relative quantitation (RQ), using the comparative CT ($2^{-\Delta\Delta CT}$) method.

**CRISPR/Cas9 KO and CRISPR/dead-Cas9 induction generation.** The desired Cas9 cutting site in the LINC01138 promoter genomic region were selected, and the target sequences in the region around 100 ~ 200 bp flanking the site were searched in http://crispr.mit.edu/. The short guide RNA (sgRNA) were designed and the sequences of high-scored sgRNAs are presented in Supplementary Data 4. The four candidate sequences (sgRNA#1-#4) were cloned into lenti-gRNA-puro. The combination of paired sgRNAs was used to expect the best efficiency. The "#1 + #3"-sgRNA pair and the "#1 + #4"-sgRNA pair were identified to be effective in both knockout system and activation system, nominated as P1 and P2, respectively. For the CRISPR-Cas9 knockout system, the lentivirus was collected 48 h after co-transfection of the lenti-gRNA1, lenti-gRNA2 and lenti-cas9-blast vector into HEK293T cells using the Lipofectamine 2000 transfection reagent (Invitrogen, Carlsbad, CA). For the CRISPR/dCas9 activation system, the co-transfected plasmids were the lenti-gRNA1, lenti-gRNA2, lenti-dCas9-vp64-blast and lenti-MS2-p65-HSF1 vector. The plasmids used in this study were purchased from Addgene (Cambridge, MA, USA). The target cells were infected with filtered lentivirus plus 6 μg/ml polybrene (Sigma-Aldrich) for 24 h and treated with 4 μg/ml puromycin (InvivoGen, San Diego, California, USA) and 4 μg/ml blasticidin (InvivoGen) for more than 7 days to obtain selective antibiotic markers before initiating the experiment. The stably transfected cells established via the CRISPR/Cas9 knockout and CRISPR/dead-Cas9 activation technology was nominated as KO and OE, respectively. These stable cell lines were all polled clones. All the cell lines were used within 20 passages and thawed fresh every 2 months.

**RNA interference and generation of lentiviral particles**. The sequences of small interfering RNA (siRNA) oligonucleotides targeting LINC01138, IGF2BP1, IGF2BP3 and PRMT5 and the negative control siRNA are provided in Supplementary Data 4; they were purchased from RiboBio (RiboBio Biotechnology, Guangzhou, China). Transfections with siRNA (75 nM) were performed with Lipofectamine 2000. The human LINC01138 sequence was cloned from SMMC-7721 cell cDNA and cloned into the BamHI and EcoRI sites of the lentivirus expression vector pWPXL to generate pWPXL-LINC01138. The IGF2BP1, IGF2BP3 and PRMT5 expression vectors were constructed by inserting the respective open-reading frame sequences into the pWPXL vector to generate pWPXL-IGF2BP1, pWPXL-IGF2BP3 and pWPXL-PRMT5, respectively. Additionally, we cloned the sequence of IGF2BP1 into the EcoRI and SpeI sites of the pCMV-FLAG vector, cloned the sequence of IGF2BP3 into the SmaI and EcoRI sites of the pCMV-FLAG vector and cloned the sequence of PRMT5 into the BamH1 and EcoRI sites of the pCMV-HA vector. The PCR primers are listed in Supplementary Data 4. The HEK293T cells were transfected with pWPXL-LINC01138, pWPXL-IGF2BP1, pWPXL-IGF2BP3 or pWPXL-PRMT5, with the packaging and envelope plasmids psPAX2 and pMD2.G, respectively (gifts from Dr. Didier), using Lipofectamine 2000 (Invitrogen) according to the manufacturer's instructions. The virus particles were collected 48 h after transfection. The HCC cells were infected with recombinant lentivirus transducing units using 1 μg/ml polybrene (Sigma-Aldrich).

**Reverse transcription PCR and quantitative real-time PCR**. RNA samples from the clinical tissue specimens and cell lines used in this study were extracted with TRIzol reagent (Invitrogen). First-strand cDNA was synthesized using the PrimeScript™ Reverse Transcriptase kit (Takara, Dalian, China). Relative RNA levels determined by quantitative Real-Time PCR (qPCR) were measured on a 7900 Real-Time PCR System with the SDS 2.3 software sequence detection system (Applied Biosystems, USA) using the SYBR Green (Takara) method. The sequences for the gene-specific primers used are listed in Supplementary Data 4. β-actin was employed as an internal control to quantify of LINC01138 and the mRNA levels of other genes. The relative levels of RNA were calculated using the comparative CT ($2^{-\Delta\Delta}$CT) method.

**RNA sequencing**. We transiently transfected $5 \times 10^6$ SMMC-7721 cells with 75 nM indicated siRNAs for 48 h, and the total RNA samples were collected by TRIzol reagent. Before the RNA libraries were constructed, rRNAs in the RNA samples were eliminated using the RiboMinus Eukaryote kit (Qiagen, Valencia, CA, USA). Next, strand-specific RNA-seq libraries were prepared using the NEBNext Ultra Directional RNA Library Prep kit (New England Biolabs, Beverly, MA, USA), according to the manufacturer's instructions[45]. Briefly, ribosome-depleted RNA samples were fragmented and prepared for first- and second-strand cDNA synthesis with random hexamer primers. The prepared cDNA fragments were treated with End-It DNA End Repair kit to repair the ends, an A was added at the 3′-end by the Klenow fragment, and finally, the fragments were ligated with adaptor sequences. The ligated cDNA products treated with uracil DNA glycosylase to remove the dUTP-labelled second-strand cDNA. The purified libraries were subjected to quality control on a Bioanalyzer 2100 (Agilent, Santa Clara, CA, USA) and sequenced using a HiSeq 3000 (Illumina, San Diego, CA, USA) on a 150-bp paired-end run. For the data processing, the raw sequencing reads were aligned to human reference genome (hg19) using the splice-aware aligner HISAT2[46]. Read counts for each gene were normalized into FPKM (Fragments Per Kilobase of transcript per Million mapped reads) values[47]. The cutoff of differential gene expression was FDR <0.05, normalized by the respective si-NC control. The mean FPKM value of the overlap genes of the two subgroups for the two independent siRNAs were used for further analysis.

**5′ and 3′ RACE assay**. We used 5′ and 3′ RACE to determine the transcriptional initiation and termination sites of LINC01138 with a SMARTer RACE cDNA Amplification kit (Clontech, California, USA), according to manufacturer's instructions. The sequences for the gene-specific PCR primers used for 5′ and 3′ RACE analysis are given in Supplementary Data 4.

**In vitro transcription and translation assay**. In vitro transcription/translation of LINC01138 was conducted using a TnT Quick Coupled Transcription/Translation Kit (Promega, USA) and detection was performed using a Transcend Non-Radioactive Translation Detection System (Promega, USA). Briefly, the TnT Quick Coupled Transcription/Translation system is completed in a single-tube and coupled transcription/translation reactions for eukaryotic cell-free protein expression of genes cloned downstream from the T7 RNA polymerase promoters. To use cell-free expression systems, 2.0 μg of circular plasmid DNA containing a T7 promoter, is added to an aliquot of the TnT Quick Master Mix and incubated in a 50 μl reaction volume for 90 min at 30 °C. The expression reaction produces significant quantities of biotinylated lysine residues proteins as detected by Transcend Non-Radioactive Translation Detection System. After SDS-PAGE and electro-blotting, the biotinylated proteins can be visualized via binding of streptavidin-horseradish peroxidase (Streptavidin-HRP), followed by chemiluminescence detection.

**Prediction of the protein-coding potency of LINC01138**. We performed a codon substitution frequency analysis using the software phyloCSF to assess the coding potential of LINC01138 in the transcripts, based on evolutionary signatures in the 29-mammalian-genome alignment[17]. In addition, CPAT (http://lilab.research.bcm.edu/cpat/), CPC (http://cpc.cbi.pku.edu.cn/) and ORF finder (https://www.ncbi.nlm.nih.gov/orffinder/) were used to predict the protein-coding potential of LINC01138. LINK-A (LINC01139) and HOTAIR served as non-coding RNA controls, and ACTB and GAPDH served as coding RNA controls.

**Northern blot**. We used a NorthernMax Kit from Ambion (Thermo Fisher Scientific, Carlsbad, California, USA) and DIG Northern starter Kit (Roche, Indianapolis, Indiana, USA) with Digoxin-labelled RNA probes to detect LINC01138 in the SMMC-7721 cells. The PCR primers of the probes of LINC01138 are listed in Supplementary Data 4. Approximately 5–10 μg of enriched polyA + RNA was loaded per lane for northern blot analysis, according to the manufacturer's instructions.

**Subcellular fractionation**. Cytoplasmic and nuclear fractions of the SMMC-7721 cells were prepared and collected according to the instructions of the Nuclear/Cytoplasmic Isolation kit (Thermo Fisher Scientific, Carlsbad, California, USA). β-actin was used as the cytoplasmic endogenous control. U2 small nuclear RNA was used as the nuclear endogenous control.

**Cell proliferation assay and colony formation assay**. The cells were seeded in 96-well flat-bottomed plates, with each well containing 1500 cells in 100 μl of cell suspension. After a certain time in culture, cell viability was measured using Cell Counting Kit-8 (CCK-8) assays (Dojindo, Kumamoto Prefecture, Japan). Each experiment with six replicates was repeated three times and measured continuously for 5 days. For colony formation assays, 1500 cells were seeded in 6-well culture dishes and allowed to grow until visible colonies formed in complete growth medium (10 days–2 weeks). Megascopic cell colonies were fixed with methanol, stained with crystal violet (Sigma-Aldrich, St. Louis, MO) and counted.

**Invasion assay and migration assay**. Invasion assays were performed in Millicell chambers in triplicate. The 8-μm pore inserts were coated with 30 μg of Matrigel (BD Biosciences, Franklin Lakes, New Jersey, USA). The migration assay was conducted similarly, without coating the filters with Matrigel. The cells ($5 \times 10^4$) were added to the coated filters in serum-free medium. We added DMEM containing 10% FBS to the lower chambers as a chemoattractant. After 24 h at 37 °C in an incubator at 5% $CO_2$, cells that migrated through the filters were fixed with methanol and stained with crystal violet. Cell numbers were counted in five random fields.

**In vivo assays**. Female athymic BALB/c nude mice, aged 4–5 weeks old, purchased from the Experimental Animal Center of Shanghai Cancer Institute (Shanghai, China). Mice (10 in each group) were injected subcutaneously with 0.2 ml of cell suspension containing $5 \times 10^5$ cells (pWPXL-VECTOR and pWPXL-LINC01138 stable SMMC-7721 cell line) in the right axilla. Tumour growth rates were monitored. When a tumour was palpable, it was measured every other day, and its volume was calculated according to the formula volume = length × width$^2$ × 0.5. Sample size was not predetermined for these experiments. To further investigate the effect of LINC01138 on tumour invasion in vivo, we developed the metastasis model in nude mice. Using the pWPXL-VECTOR and pWPXL-LINC01138 stable SMMC-7721 cell lines, 40 μl matrigel containing $1 \times 10^6$ cells were injected into the liver, and the mice were killed after 9 weeks. The number of metastatic foci in the liver, lung and intestine were determined using the hematoxylin eosin (H&E) (Beyotime Biotechnology) staining in tissue sections under a binocular microscope (Leica, Wetzlar Lottehaus, Germany). All experiments were performed in accordance with relevant institutional and national guidelines and regulations of Shanghai Medical Experimental Animal Care Commission.

**RNA pull-down assays**. LINC01138 or antisense-LINC01138 RNAs were transcribed and labelled by the Biotin RNA Labeling Mix (Roche, USA), treated with RNase-free DNase I (Takara, Japan) and purified with an RNeasy Mini Kit (QIAGEN, USA). Next, 1 pmol biotinylated RNA was pretreated with RNA structure buffer (Beyotime Biotechnology, Shanghai, China) to obtain an appropriate secondary structure formation. The pre-treated biotinylated RNAs were incubated with 1 mg protein extracts of SMMC-7721 cells at 4 °C for 1 h, gently mixed with 40 μl washed streptavidin beads (Invitrogen, USA) and incubated on a rotator overnight. The beads were washed briefly five times in 1× washing buffer (5 mM Tris-HCl, 1 M NaCl, 0.5 mM EDTA, and 0.005% Tween 20). The proteins were precipitated and diluted in 60 μl protein lysis buffer, separated by gel electrophoresis and visualized by silver staining. Specific bands were excised for proteomics screening by mass spectrometry analysis (Shanghai Applied Protein Technology, Shanghai, China). Protein identification was retrieved in the human RefSeq protein database (National Center for Biotechnology Information), using Mascot version 2.4.01 (Matrix Science, London, UK). The retrieved protein was detected by western blot. The primers of LINC01138 and its deletion fragments for

in vitro transcription are provided in Supplementary Data 4. The uncropped scan of gels is shown in Supplementary Fig.14.

**RNA immunoprecipitation (RIP) assay**. RIP assays were performed using the Magna RIP RNA-binding protein immunoprecipitation kit (Millipore, Massachusetts, USA). Briefly, cells growing in 10 cm-dishes were lysed in 0.5 ml of lysis buffer containing protease inhibitors and RNase Inhibitor (Thermo Fisher Scientific Inc., Rockford, IL, USA) and centrifuged at 12,000 r.p.m. for 30 min. The supernatants were incubated with Protein G Dynabeads (Thermo Fisher Scientific, Carlsbad, California, USA), which were incubated with the indicated antibodies for 12 h at 4 °C with gentle rotation. The beads were washed thrice with wash buffer containing RNase inhibitor and then twice with PBS containing RNase inhibitor. The RNA was extracted using the Total RNA isolation kit (Thermo Fisher Scientific Inc), and qRT-PCR was performed as described above. For the RIP assays of deletion mutants, 5 pmol plasmids with FLAG- or HA-tagged full-length and truncated IGF2BP1, IGF2BP3 or PRMT5 were transiently transfected into the SMMC-7721 cells, and the cell lysates were immunoprecipitated with the indicated antibodies. Information on the antibodies are listed in Supplementary Table 3.

**Immunoblotting analysis**. Cells ($5 \times 10^6$) were lysed for 20 min with lysis buffer (Beyotime Biotechnology) containing protease inhibitors (Roche, Indianapolis, IN, USA). The protein concentrations were determined by the BCA method (Pierce, Thermo Fisher Scientific Inc., Rockford, IL, USA). After centrifugation at $16,400 \times g$ for 15 min at 4 °C, the samples were resolved by SDS/PAGE, transferred to PVDF membranes (Immobilon-P membrane, Millipore, Massachusetts, USA), and analysed by immune blotting using HRP-conjugated secondary antibodies. The membranes were blocked with 5% (wt/vol) skimmed milk in TBS plus Tween 20 at 4 °C overnight before probing with antibodies. Information on the antibodies are provided in Supplementary Table 3. An enhanced chemiluminescent (ECL) chromogenic substrate was used to visualize the bands (Pierce). Visualization was performed using the Enhanced Chemiluminescence Plus Western Blotting Detection System (GE Healthcare, Connecticut, USA) and LAS-4000EPUV mini Luminescent Image Analyzer (GE Healthcare). The uncropped blots are shown in Supplementary Fig. 14.

**Immunoprecipitation assay**. The indicated plasmids were transfected into SMMC-7721 cells that had been treated with or without the proteasome inhibitor MG132. The cells were lysed in RIPA buffer (Beyotime Biotechnology) containing protease inhibitors and RNase Inhibitor (Life Technologies) and centrifuged at $16,400 \times g$ for 15 min. The supernatants were incubated with anti-FLAG or anti-HA Protein G Dynabeads (Life Technologies) overnight at 4 °C with gentle rotation. The beads were washed thrice with NT2 buffer (50 mM Tris-HCl, pH 7.4, 150 mM NaCl, 0.05% Nonidet P-40, 1 mM MgCl$_2$) containing protease inhibitors (Thermo Fisher Scientific Inc.) and RNase Inhibitor (Thermo Fisher Scientific Inc.) and twice with PBS containing protease inhibitors and RNase Inhibitor (Thermo Fisher Scientific Inc.). After washing, the proteins were eluted by competition with FLAG or HA peptides (Thermo Fisher Scientific Inc.). The immuno-complexes were analysed by SDS/PAGE and immunoblotting with anti-Flag, anti-HA, anti-IGF2BP1, anti-IGF2BP3 or anti-PRMT5 antibody.

**In vitro cellular IC$_{50}$ assays**. Vector and pWPXL-LINC01138 SMMC-7721 cells were seeded in 96-well flat-bottomed plates, with each well containing 5000 cells in 100 μl of cell suspension to determine the concentration that causes 50% inhibition of cell viability. According to the recommended concentrations of each inhibitors, we performed 10 or 20 concentration gradients. After 48 h in culture, the cell viability was measured using Cell Counting Kit-8 (CCK-8) assays (Dojindo). Each experiment with six replicates was repeated three times.

**Statistical analysis**. Data were presented as the mean ± standard error of the mean (SEM) from one representative experiment of five independent experiments. Unless stated otherwise, Student's $t$-test or one way analysis of variance (ANOVA) were performed to evaluate the differences between two groups or more than two groups, respectively, followed by Dunnett's multiple comparisons test. Wilcoxon tests were used to analyse LINC01138 RNA levels in paired human samples, and Mann–Whitney tests were used to analyse LINC01138 RNA levels in grouped human samples. The Cox univariate proportional hazards regression models was used to determine the independent clinical factors based on the investigated variables. The long-rank Mantel-Cox test was performed to analyse the correlation between the *LINC01138* RNA levels and the overall survival. The $\chi^2$-test was performed to analyse the function of LINC01138 on HCC cell metastasis in vivo. Pearson's correlation was performed to analyse the correlation of the gene expression profiling between the si-LINC01138 group and si-PRMT5 group. $P$-value of $<0.05$ was considered statistically significant. All statistical analyses were performed using the IBM SPSS Statistics V19 package (Armonk, NY, USA).

**Data availability**. The RNA sequencing data from this study have been deposited in the Gene Expression Omnibus (GEO, 'GSE111655'). The authors declare that all relevant data of this study are available within the article or from the corresponding author on reasonable request.

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

## Acknowledgements

We are grateful for Dr T. Didier's gifts of the pWPXL, psPAX2 and pMD2.G lentivirus plasmids. We also much appreciate Professor Jingxuan Pan for providing us the PRMT5 inhibitor (PJ-68). This work was supported by grants from the National Natural Science Foundation of China (81672774, 81790252, 81301712, 81672727 and 81522036). We also thank to the supports from Foundation of Key Laboratory of Gene Engineering of the Ministry of Education (201501) and Foundation of State Key Laboratory of Oncogenes and Related Genes (90-16-01).

## Author contributions

X.H., Y.Z. and Z.L. designed the study; Z.L., J.Z., X.L., Z.H., J.D. and T.Y. performed the experiments; Z.L., J.Z., X.L., Q.G., Y.Z. and X.H. analyzed the results; X.L., D.C. and Q.W. performed IHC; J.F, J.L. and Q.G. provided the HCC samples; M.Y. performed animal experiments; S.H. and S.L. carried out bioinformatics analyses; Z.L., Y.Z. and X.H. wrote the paper with comments from all authors.

## Additional information

**Competing interests:** The authors declare no competing interests.

