## [Peer Review File · Nature Communications]

Reviewers' comments:

Reviewer #1 (Remarks to the Author):

Comments on Li et al. The LincRNA OncLn1 Drives Malignancies via Activating Arginine Methyltransferase 5 in Hepatocellular Carcinoma

Based on RNA-seq and genomic amplification data, the authors identify lincRNAs deregulated in HCC and aim to decipher the molecular mechanism for Linc01138, which they rename to OncLn1. In short, the authors propose that Linc01138 is stabilized by IGF2BP1/3 in the cytoplasm, and binds and stabilizes the methyltransferase PRMT5 by preventing the association of PRMT5 to the E3 ubiquitin ligase CHIP. This makes biological sense as PRMT5 is known to have oncogenic functions.

Although the manuscript is interesting and at times comprehensive, most of the experiments are very poorly described and there appear to be logical flaws in the argumentation as well.

Specific points.

Figure 1:

The authors overlay TCGA expression data with a study of copy number analysis in HCC using only a small number of patients. From this analysis LINC01138 is selected for further studies.

- Why did the authors not use TCGA data for the copy number analysis? According to the TCGA, LINC01138 is not amplified in liver cancer (>1100 samples). How do the authors explain this discrepancy?
- This reviewer strongly opposes renaming lincRNAs, as it frequently leads to duplicate publications on the same gene and clouds the literature. The official symbol is LINC01138.
- Regarding the copy number analysis: According to Supplementary table 1 only 26 samples were included in the analyses for LINC01138. Please clarify. The legends and Materials and Methods sections are inadequate. I have doubts if this analysis was done correctly as hardly any patients appear to be diploid at this locus - even in the NT samples.
- Is there a correlation between gene amplification and Linc expression?
- How were the patients stratified for the analysis in Figure 1g?

Figure S2:

- The northern blot in fig S2d indicates 2 RNA species that are considerably longer than 2075 nts (not bp as stated in the manuscript). The northern blot should be repeated with siRNAs placed in various exons and featuring samples from the CRISPR activation and KO samples.
- Fig S2c displays a 3-exon transcript but the 2075 nts transcript appears to have 4 exons. Please clarify.
- The bioinformatics tools used to analyse coding potential indicate (not demonstrate) that linc01138 is non-coding. Does LINC01138 associate with the ribosome?

Figure 2:

- The CRISPR experiments are very poorly described. The authors refer back to a previous paper but there the description is equally poor. It is unclear if these are transient or stable assays? Only HEK293 cells are mentioned in the methods section and here the experiment appears to be transient. Other cells are used in the results section. Where the experiments performed on a cell population or were clones selected? As the authors have derived CRISPR cell lines for LINC01138 overexpression and ablation, these should be used in most of the following experiments.
- The CRISPR KO cells should be used in the in vivo experiments to demonstrate a physiologically relevant role for LINC01138 in tumor outgrowth.

Figure 3:

- The RNA pulldown/MS experiments are very poorly described. How much RNA was used? How

were the precipitates washed?

- Supplementary Table 3 should include all 3 MS experiments AND include the number of peptides retrieved with the negative control antisense RNA. It would also be helpful to state the approx. MW of the proteins.
- The experiments in Fig 3b are lacking controls for specificity.
- The experiments in Fig 3d have no negative control for specificity (GAPDH or the like)
- Figure 3e,f: The authors state that the C-term of IGF2BP1 is responsible for binding LINC01138. I think the authors mean N-terminal? Still the interpretation is not correct as these domains alone cannot bind the lincRNA.

Figure 4:

- Figure 4a-c: These experiments should be performed in the CRISPR KO and OE cells. Also, the degree of lincRNA modulation should be displayed.
- Figure 4d: This does not look like ubiquitination? What are the MW of these bands? Analyses of the endogenous PRMT5 should be performed. The CRISPR cells should be employed.
- Figure 4e,f: The CRISPR cells should be employed. Alternatively, 2 independent siRNAs should be used. All panels should display both PRMT5 and CHIP expression.

Figure 5:

- Figure 5a: The effect of siRNA-PRMT5 alone should be displayed (moved from suppl.). The observed effects may be independent from the functions of the lincRNA. The interesting (and necessary) experiment would be to assay whether an overexpression of PRMT5 can rescue the effects of knocking out the LINC01138.
- Figure 5d: There is barely any methods description for this experiment, so it is unclear how the experiment was performed and the data analysed. According to the supplementary data, two independent siRNA were used for both the lincRNA and PRMT5? At which concentrations and at what time point? What is actually displayed in the figures - the overlap between the two siRNAs? What the same negative control siRNA used in both experiments? Could off-targets of the control siRNA be driving the comparison between the two experiments?
- What is the actual overlap in affected transcripts between the lincRNA and PRMT5 siRNA experiments?
- Figure 5f,g: The authors should explore the importance of the lincRNA for PRMT5 regulated genes. Can PRMT5 overexpression rescue a knockdown or knockout of the lincRNA?

Figure 6:

- The authors must demonstrate how an active site-binding small drug can inhibit the association to the lincRNA, which the authors claim prevents binding of PRMT5 to CHIP.
- Figure 6d: How do the authors envision this to happen? To this reviewer, an additive effect of siRNA to LINC01138 and the PRMT5 drugs would suggest that they act in different, rather than in the same, pathway.

Reviewer #2 (Remarks to the Author):

In this manuscript, Li et al. characterized an oncogenic Long non-coding RNA OncLn1, which is frequently amplified and overexpressed in Hepatocellular carcinoma (HCC). The authors identified OncLn1 through analysis of RNA-seq data of HCC and adjacent tissues. They also confirmed the tumor-promoting function of OncLn1 in HCC by either overexpression or knockout/knockdown of OncLn1 in HCC cells. Subsequently they performed OncLn1 RNA pull-down assay and mass spectrometry, from which they found out IGF2BP1, IGF2BP3 and PRMT5 are binding partners of OncLn1 in HCC cells. The authors showed OncLn1 binds to PRMT5 and protects PRMT5 from being degraded by proteasome. In addition, PRMT5 mediates OncLn1-induced HCC cell growth, migration and invasion. While IGF2BP1 and IGF2BP3 act upstream of OncLn1 and enhance RNA stability of OncLn1. In conclusion, the discovery of OncLn1 and OncLn1/PRMT5 axis can be a therapeutic

target of HCC.

The authors did many experiments by engaging gain-of-function and loss-of-function approaches to confirm their hypotheses. Most of them are well supported, however, some of the data were not correctly interpreted. Meanwhile, the manuscript should be written carefully and there are several obvious grammar errors in the text. Also, the paper is not readable and hard to understand for non-experts in the field because it lacks of explanations for many treatments/means used in the experiments. Similarly, the authors did not describe their experiments/results clearly either in the text or in the figure legend. The authors should carefully go through the manuscript and rewrite the text to make it more accessible to readers. Here are some specific points need to be addressed/explained as described below in details.

1. There is no Supplementary Fig 2j in the supplementary data so the citations of the figures in the page 7 are not right. The results are not compatible with the figures, which makes it hard to understand. Those highlighted text below should be corrected.

HCC cell lines and HCC tissues (Supplementary Fig. 2b-f). In addition, the PhyloCSF codon substitution frequency analysis 17, Coding Potential Assessment Tool (CPAT), Coding Potential Calculator (CPC) <http://cpc.cbi.pku.edu.cn/> and ORF finder software from the National Centre for Biotechnology Information (NCBI) demonstrated that OncLn1 is a bona fide long noncoding RNA (Supplementary Fig. 2g, h). Moreover, OncLn1 is widely expressed in different liver cancer cell lines (Supplementary Fig. 2i) and distributes in both the cytoplasm and nucleus of SMMC-7721, SNU-449 and Huh-7 cells (Supplementary Fig. 2j).

2. In Fig 2, what do ctrl-cas9-VP64, OncLn1- (1+3) and OncLn1- (1+4) mean? Although as professionals, we may know the first one is control and the other two are overexpression of OncLn1 in HCC cells. At least it should be explained in the results, figure legends or method part. Similarly, what do OncLn1-sg (1+3) and OncLn1-sg (1+4) mean? The authors should include more detailed information about the experiment design rather than just introduce the experiment name in figure legend.

3. OncLn1 RNA level achieved by overexpression of lentivirus is much higher (almost 100 folds) than endogenous activation, while the colony formation assay showed that the endogenous activation (Fig2a) indeed had stronger colony formation ability than letiviral overexpression (Supplementary Fig3f). The colony formation ability of the control cells as well as the parental (Mock) cells are similar according to the figures. Why high OncLn1 level do not achieve stronger colony formation ability?

4. In Fig 3c, what are the exact meanings of red boxes. Are they match with the OncLn1 deletions? It is confusing.

5. As is described in the results, 220-630nt fragment of OncLn1 mediates interaction with IGF2BP1 or IGF2BP3. However, in Fig 3d, there is a clear band of IGF2BP1 in #5 (630-2075nt) RNA pull-down complex, while there is no IGF2BP1 or IGF2BP3 in #7 (220-2075nt) RNA pull-down complex. Is that because of the distinguished RNA structures of #5 and #7? Do you have evidence? Otherwise it is not correct to conclude that 220-630nt fragment of OncLn1 mediates its interaction with IGF2BP1 and IGF2BP3.

6. In page 9, text showing below is confusing.

Next, RIP assay results showed that the RNA recognition motif (RRM) and K homology RNA binding domain (KH) at the C-terminus of IGF2BP1 and IGF2BP3 (Fig. 3e, f), mediated their interactions with OncLn1, whereas the S-adenosylmethionine (SAM)-binding domain at the N-terminus of PRMT5 physically associates with OncLn1 in the HCC cells (Fig. 3g).

What does it mean? It is obvious that in Fig 3e and 3f, RRM alone is not efficient to pull down OncLn1. SAM domain (350-440 aa) as showing in Fig 3g is not in the N-terminal of PRMT5 (637aa). There is no significance analysis showing in the figure, readers will unable to tell what

exactly the authors want to say. Is 5 folds significant or 10 folds significant compare with IgG control? It will be different interpretations regarding different definition of significance.

7. Many labels of western blot figures are confusing. For example, in Fig 4e and 4f, what does the IgG do here? The figure labels showed that these experiments used FLAG or HA antibodies for immunoprecipitation, so IgG was also used for immunoprecipitation? Alternatively, IgG means a plasmid used here as a control of pCMV-FLAG-CHIP/ pCMV-HA-PRMT? In addition, there is no explanation about it in figure legend. There are lots of mixed use of plasmid names and protein names for western blot labels. The authors should check carefully and try to make it clear and correct.

Reviewer #3 (Remarks to the Author):

Through analyzing TCGA database, Li et al discovered that LINC01138 (they named it LincRNA OncLn1 in this paper) was significantly increased in HCC tissues which was associated with the poor outcome of HCC patients. They further studied the biological functions and mechanisms of OncLn1 in HCC tumorigenesis. The authors found that OncLn1 exerted oncogenic functions in HCC tumorigenesis through promoting HCC cell growth and metastasis. RNA pull-down assay showed that IGF2BP1, IGF2BP3 and PRMT5 physically associate with OncLn1. The authors further showed that OncLn1 increased PRMT5 stability through blocking its ubiquitination.

This study is potentially interesting, however LINC01138, as a novel oncogenic long intergenic non-coding RNA (lincRNA), promoted proliferation with anti-apoptosis activity in prostate cancer cells, also as a potential biomarkers, has just been reported recently from the same city, which thus spoils the novelty of the current study.

Some of their data were over-interpreted, and several key scientific questions have not been addressed adequately. The conclusion of oncogenic IGF2BP1/IGF2BP3-OncLn1-PRMT5 axis in HCC still lacks solid experimental evidence.

Comments:

1. The authors found that OncLn1 upregulates PRMT5 expression through inferring it binding with E3 ligase CHIP and further blocking its degradation. As substrate specificity of proteasome-dependent degradation, it seems that E3 ligase CHIP may be a key regulator in OncLn1-PRMT5 axis-mediated oncogenic behavior. However, the authors didn't show the expression levels of CHIP in HCC tumor samples, and its regulation by OncLn1.

2. As no effects were observed on expression levels of IGF2BP1 and IGF2BP3 by OncLn1 in HCC cells, the authors showed that IGF2BP1 and IGF2BP3 could be important upstream effectors of OncLn1 in HCC cells through enhancing OncLn1 stability. However, the authors didn't find key regulators mediating stabilization of OncLn1 by IGF2BP1 and IGF2BP3, as IGF2BP1 and IGF2BP3 should be adaptor proteins.

Point-by-point responses to the reviewers' comments

NCOMMS-17-10603A: “The LincRNA OncLn1 Drives Malignancies via Activating Arginine Methyltransferase 5 in Hepatocellular Carcinoma”.

Reviewer #1 (Remarks to the Author):

Comments on Li et al. The LincRNA OncLn1 Drives Malignancies via Activating Arginine Methyltransferase 5 in Hepatocellular Carcinoma

Based on RNA-seq and genomic amplification data, the authors identify lincRNAs deregulated in HCC and aim to decipher the molecular mechanism for Linc01138, which they rename to OncLn1. In short, the authors propose that Linc01138 is stabilized by IGF2BP1/3 in the cytoplasm, and binds and stabilizes the methyltransferase PRMT5 by preventing the association of PRMT5 to the E3 ubiquitin ligase CHIP. This makes biological sense as PRMT5 is known to have oncogenic functions.

Although the manuscript is interesting and at times comprehensive, most of the experiments are very poorly described and there appear to be logical flaws in the argumentation as well.

Specific points.

Figure 1:

The authors overlay TCGA expression data with a study of copy number analysis in HCC using only a small number of patients. From this analysis LINC01138 is selected for further studies.

1. - Why did the authors not use TCGA data for the copy number analysis? According to the TCGA, LINC01138 is not amplified in liver cancer (>1100 samples). How do the authors explain this discrepancy?

Response: Thanks for the Reviewer's professional comments. LINC01138 is one of the lincRNAs with copy number gains in HCC (**the revised Supplementary Table 1**), based on the SNP assay data which has been reported in our previous finding (Jia D et al., Hepatology, 2011). In the previous SNP array, DNA samples from the 58 paired primary HCC tumors and the adjacent non-tumor tissues were amplified and hybridized onto an Affymetrix Genome-Wide human SNP array 6.0 (Affymetrix Inc.). The overall hybridization quality was estimated by the call rate index obtained from Genotyping Console Software (GTC 3.0, birdseed algorithm using default parameter settings). **Copy number was calculated based on probe hybridization signal intensity data relative to the signal in adjacent non-tumor tissue from the same patient. Regions of CNAs were detected using a Hidden Markov Model (HMM) algorithm in the standard Partek workflow for paired samples.** Regions were detected using a minimum of 7 probe sets, and the criteria to define significant CNAs were as follows: a gain or loss was inferred for a relative copy number of greater than 2.6 or less than 1.3, respectively; the minimum physical length of the putative CNA was more than 100 kb; the CNA was present in at least three tumor samples; and finally, the overlapping common regions among multiple tumors were calculated.

Furthermore, Gene annotation and overlap were carried out using the University of California, Santa Cruz (UCSC) hg38, to assemble the lincRNAs in the regions with significant genomic gains. and 53 candidate lincRNAs were selected out according to the criteria (relative CNAs in >30% HCC samples, occurred in Amplication CNA area, prior to long intergenic non-coding RNA; **the revised Supplementary Table 1**). With the intersection of these 53 candidate lincRNAs with HCC genomic gains and the 1,082 lincRNAs up-regulated in the TCGA cohort of 50 paired HCC tissues and adjacent

non-tumor (NT) tissues, four lincRNAs arised, including *Linc01138*, *PVT1*, *RP11-14N7.2* and *RP11-30J20.1* (the revised **Fig. 1a** and **Supplementary Fig. 1a, b**). Among them, LINC01138 (ENSG00000274020.1) showed the strongest mean signal intensity of CNAs (40.6367425 in HCC tissues, 0 in the adjacent NT tissues). **Real-time qPCR assays confirmed that Linc01138 showed more genomic gains in HCC tissues, compared with their corresponding adjacent NT liver tissues, in two independent cohorts (the revised Fig. 1b and 1c, Cohort 1, 72 paired patients HCC and corresponding adjacent NT liver tissues obtained from the surgical specimen archives of the First Affiliated Hospital of Zhejiang University; the revised Fig. 1d and 1e, Cohort 2, 120 paired patients HCC and corresponding adjacent NT liver tissues obtained from the surgical specimen archives of the Zhongshan Hospital of Fudan University). The related information has been added in the revised Methods and Results (page 6 in the revised version).**

Actually, We did have analyzed the copy number data of Linc01138 in a TCGA cohort, which contains the available CNV data of **374 HCC tissues**, and found that this lincRNA did not show obvious absolute amplification signals in these HCC tissues. However, The CNV data for the corresponding adjacent non-tumor liver tissues is not available, so it is difficult to assess whether Linc01138 has obvious genomic gains in the these HCC tissues relative to the paired adjacent non-tumor tissues. Moreover, we evaluated the copy numbers of the 53 candidate lincRNAs in the TCGA-374-cohort, and surprisingly found that only 7 lincRNAs (ENSG00000249859.6, ENSG00000232527.6, ENSG00000254101.4, ENSG00000237343.1, ENSG00000236140.1, ENSG00000253227.1, ENSG00000247317.3) showed significant absolute genomic gains in these HCC tissues. But interestingly, our pilot experiments revealed that both the lincRNAs among these 7 linRNAs (ENSG00000232527.6, eg.; **Fig. R1A, B**) and the rest 46 lincRNAs (ENSG00000230623.2, eg.; **Fig. R1C, D**) have increased copy numbers in HCC tissues, compared with their corresponding adjacent NT tissues. These results demonstrated that our previous SNP data from the 58 paired primary HCC tumors and the adjacent non-tumor tissues could give additional informations to the CNV data of the TCGA-

374 cohort.

Figure R1. Copy number analysis for ENSG00000232527.6 (A and B) and ENSG00000230623.2 (C and D) in DNA samples from 40 paired primary HCC tumors and the adjacent non-tumor tissues, using qPCR.

2. - This reviewer strongly opposes renaming lincRNAs, as it frequently leads to duplicate publications on the same gene and clouds the literature. The official symbol is LINC01138.

Response: We will follow the reviewer's suggestion, and use "LINC01138" instead in the revised manuscript.

3. - Regarding the copy number analysis: According to Supplementary table 1 only 26 samples were included in the analyses for LINC01138. Please clarify. The legends and Materials and Methods sections are inadequate. I have doubts if this analysis was done correctly as hardly any patients appear to be diploid at this locus - even in the NT samples.

Response: Thanks for the Reviewer's professional comments. Based on the previous SNP assay data of 58 paired primary HCC tumors and the adjacent non-tumor tissues (Jia D et al., Hepatology, 2011), LINC01138 was considered to be amplified in the HCC tissues of 26 patients, for about 45% in the total 58 HCC patients, comparing with the corresponding adjacent non-tumor tissues. **We have modified the description in the revised Supplementary Table 1 to be "CNAs Frequency (% in Samples): 45% (26/58)" for better understanding. According to Reviewer's advice, we have added the related information of "Genomic DNA copy number alteration analysis" in the revised Materials and Methods (in the revised Supplementary data), as follows:**

"A SNP array data (Affymetrix Genome-Wide Human SNP Array 6.0) of 58 paired HCC and non-tumor tissues was used to predict the gain or loss for each gene, inferred for a relative copy number.¹ In brief, copy number was calculated based on the hybridization signal intensity, compared to the signal in adjacent non-tumor tissue from the same patient. Significant CNAs were detected using a Hidden Markov Model (HMM) algorithm in the standard Partek workflow for paired samples, with the criteria as follows: the minimum physical length of the putative CNA was no less than 100 kb; a gain or loss was inferred for a relative copy number of greater than 2.6 or less than 1.3, respectively; the CNA was present in at least three tumor samples; and the overlapping common regions among multiple tumors were calculated. Gene annotation and overlap were carried out using the University of California, Santa Cruz (UCSC) hg38, to assemble the lncRNAs in the regions with significant genomic gains. Real-time qPCR was performed to determine the relative copy number of the target genes in paired HCC tissues and the corresponding adjacent non-tumor tissues, using SYBR Green (Takara, Japan) in the 7900HT Fast Real-Time PCR System (Applied Biosystems, USA). The number of copies of the target genes in each test sample is

determined by relative quantitation (RQ), using the comparative CT ($2^{-\Delta\Delta CT}$) method.”

4. - Is there a correlation between gene amplification and Linc expression?

Response: Thanks a lot for the Reviewer’s good suggestion. **As shown in Figure R2,** There was a positive correlation between the genomic copy number and the RNA level of LINC01138 in a 160-patient cohort ($r = 0.3466$, $P = 0.0002$). A total of 160 patients from the Zhongshan Hospital of Fudan University, whose DNA samples and RNA samples were both available, were randomly selected from HCC patients underwent primary and curative resection in 2013-2017, and enrolled in this study for correlation analyse.

Figure R2. The correlation between gene amplification and RNA expression levels of LINC01138 in a 160-patient cohort.

5. - How were the patients stratified for the analysis in Figure 1g?

Response: Thanks for the Reviewer’s comment. In the original version, the 120-patient cohort were stratified into two groups according to the median value of LINC01138 RNA level. Kaplan–Meier overall survival analysis revealed that patents with low LINC01138 expression had longer median overall survival (OS) (38.6 versus 23.7 months, $P=0.01$, log-rank test, **the revised Figure 1g**). **We have added the related information in the revised Figure Legends (page 22 in the revised version), as “Patients were stratified for the analysis by the median value”.**

Figure S2:

6. - The northern blot in fig S2d indicates 2 RNA species that are considerably longer than 2075 nts (not bp as stated in the manuscript). The northern blot should be repeated with siRNAs placed in various exons and featuring samples from the CRISPR activation and KO samples.

Response: We appreciate the Reviewer's advices. As kindly suggested by the reviewer, we repeated northern blot in RNA samples from the CRISPR activation or CRISPR knockout cells. As shown in the revised Supplementary Figure 2d, the CRISPR system could effectively activated or knockout the endogenous LINC01138, and the 2075-nt LINC01138 variant is the predominant transcript in these cells. **We have provided the updated results in the revised Supplementary Figure 2d and stated the predominant variant as "2075-nt LINC01138" instead in the revised version (page 6 in the revised version).**

7. - Fig S2C displays a 3-exon transcript but the 2075 nts transcript appears to have 4 exons. Please clarify.

Response: Thanks for the Reviewer's comment. The red letters in the original version represented the 5' end and 3' end sequence identified by RACE assays, not the different exons. **We have improved the labels and provided related annotation in the revised Supplementary Figure 2c, and updated the related figure legend as "The sequence identified by RACE assays are underlined. The boundary of exons are shown as bold and italic."** .

8. - The bioinformatics tools used to analyse coding potential indicate (not demonstrate) that linc01138 is non-coding. Does LINC01138 associate with the ribosome?

Response: Thanks for the reviewer's comment. In the revised version, LINK-A (LINC01139) and HOTAIR served as non-coding RNA controls, and ACTB and GAPDH served as coding RNA controls, when the coding potential of the 2705-nt LINC01138 transcript was analysed by bioinformatics tools. **The bioinformatic**

prediction strongly suggested that LINC01138 is a non-coding RNA (the revised Supplementary Figure 3a-d). Moreover, we carried out *In vitro* transcription and translation assay to determine the coding potential of this lincRNA. In brief, LINC01138 was cloned into a circular plasmid DNA with the T7 RNA polymerase promoter upstream, using Luciferase gene as coding RNA control. Then, coupled transcription/translation reactions for eukaryotic cell-free protein expression of LINC01138 were provided convenient single-tube, according to the manufacturer's instructions. The *in vitro* synthetic protein products contained biotinylated lysine residues and could be detected by Transcend Non-Radioactive Translation Detection System. **As shown in the revised Supplementary Figure 3e, LINC01138 RNA could be successfully transcribed (the left), but failed to be translated into protein (the right).** Though there is no direct evidence that LINC01138 does not associated with the ribosome, we can confirm that LINC01138 is a non-coding RNA. **We have provided the statements in the revised Results (page 7 in the revised version), and added the related information of *In vitro* transcription and translation assay in the revised Methods.**

Figure 2:

9. - The CRISPR experiments are very poorly described. The authors refer back to a previous paper but there the description is equally poor. It is unclear if these are transient or stable assays? Only HEK293 cells are mentioned in the methods section and here the experiment appears to be transient. Other cells are used in the results section.

Response: Thanks for the Reviewer's professional comments. **The related informations of CRISPR/Cas9 and CRISPR/dead-Cas9 technology have been provided in the revised Methods for better readability, as follows:**

“The desired Cas9 cutting site in the LINC01138 promoter genomic region were selected, and the target sequences in the region around 100bp~200bp flanking the site were searched in <http://crispr.mit.edu/>. The short guide RNA (sgRNA) were designed and the sequences of high-scored sgRNAs are presented in **Supplementary Table 6.**

The four candidate sequences (sgRNA#1-#4) were cloned into lenti-gRNA-puro, respectively. The combination of paired sgRNAs was used to expect the best efficiency. The “#1+#3”-sgRNA pair and the “#1+#4”-sgRNA pair were identified to be efficive in both knockout system and activation system, nominated as P1 and P2, respectively. For CRISPR-Cas9 knockout system, Lenti-virus was harvested 48 hours after co-transfection of the lenti-gRNA1, lenti-gRNA2 and lenti-cas9-blast vector into HEK-293T cells using Lipofectamine 2000 transfection reagent (Invitrogen, Carlsbad, CA). For CRISPR/dCas9 activation system, the co-transfected plasmids were the lenti-gRNA1, lenti-gRNA2, lenti-dCas9-vp64-blast vector and lenti-MS2-p65-HSF1 vector. The plasmids used in this study have purchased from Addgene (Cambridge, MA, USA). Target cells were infected with filtered lenti-virus plus 6 µg/mL polybrene (Sigma-Aldrich) for 24 hours, then treated with 4 µg/mL puromycin (InvivoGen, San Diego, California, USA) and 4 µg/mL blasticidin (InvivoGen) for more than 7 days to get selective antibiotic markers before the following manufacture. The stably cells established via the CRISPR/Cas9 knockout and CRISPR/dead-Cas9 activation technology was nominated as KO and OE, respectively. These stable cell lines were all polled clones. All the cell lines were used within 20 passages and thawed fresh every 2 months.”

HEK-293T cells were transiently transfected to harvest lenti-virus. Target cells were infected by the indicated virus and selected by the indicated antibiotics to generate stable cell lines. These stable cell lines were all polled clones. All the cell lines were used within 20 passages and thawed fresh every 2 months. We have added these informations into the revised Methods.

10. Where the experiments performed on a cell population or were clones selected? As the authors have derived CRISPR cell lines for LINC01138 overexpression and ablation, these should be used in most of the following experiments.

Response: We appreciate the Reviewer’s kind advices. The CRISPR cell lines were generated after infection by the indicated lenti-virus and selection of the indicated antibiotics. These stable cell lines were all polled clones. All the cell lines were used within 20 passages and thawed fresh every 2 months. **We have provided the information in the revised Methods.**

11. - The CRISPR KO cells should be used in the in vivo experiments to demonstrate a physiologically relevant role for LINC01138 in tumor outgrowth.

Response: As kindly advised by the Reviewer, we evaluated the performance of CRISPR-LINC01138-KO cells in the *in vivo* assays. 1×10^6 CRISPR-VECTOR cells or CRISPR-LINC01138-KO cells were suspended in 40 μ l matrigel and injected orthotopically into the left hepatic lobe of each mouse. The mice were sacrificed after 9 weeks, and the number of metastatic foci in liver, lung and intestine were determined in Hematoxylin Eosin staining in tissue sections. As expected, the metastatic nodules were significantly reduced in the CRISPR-LINC01138-KO-P1 group (**the revised Supplementary Fig. 6f**). Not surprisingly, Haematoxylin-Eosin staining showed that the metastatic foci derived from CRISPR-LINC01138-KO-P1 cells dramatically decreased in the liver and lung sections (**the revised Supplementary Fig. 6g and 6h**), indicating the oncogenic effect of LINC01138. These results strongly support our previous results of the *in vivo* assays base on pWPXL-LINC01138 cells, and demonstrated that LINC01138 acts as an oncogenic driver in the development and progression of HCC. **We have added the detailed data in the revised Results (page 8 in the revised version).**

Figure 3:

12. - The RNA pulldown/MS experiments are very poorly described. How much RNA was used? How were the precipitates washed?

Response: Thanks for the Reviewer's professional comments. We used 1 pmol biotinylated RNA in the incubation system, and the beads-RNA-proteins were then washed with $1 \times$ binding washing buffer (5 mM Tris-HCl, 1 M NaCl, 0.5 mM EDTA, and 0.005 % Tween 20) for five times. **And we have improved the related informations of RNA Pull-Down/MS Assays in the revised Methods, as follows:**

“LINC01138 or antisense-LINC01138 RNAs were transcribed and labelled by the Biotin RNA Labeling Mix (Roche, USA), then treated with RNase-free DNase I (Takara, Japan) and purified with RNeasy Mini Kit (QIAGEN, USA). 1 pmol

biotinylated RNA was pretreated with RNA structure buffer (Beyotime Biotechnology, Shanghai, China) to get proper secondary structure formation. The pretreated biotinylated RNAs was incubated with 1 mg protein extracts of SMMC-7721 cells at 4°C for one hour, then gently mixed with 40µL washed streptavidin beads (Invitrogen, USA) and incubated on a rotator overnight. Then, Beads were washed briefly for five times in 1×washing buffer (5 mM Tris-HCl, 1 M NaCl, 0.5 mM EDTA, and 0.005 % Tween 20). The proteins were precipitated and diluted in 60µl protein lysis buffer, then separated by gel electrophoresis and visualized by silver staining. Specific bands were excised for proteomics screening by mass spectrometry analysis (Shanghai Applied Protein Technology, Shanghai, China). Protein identification was retrieved in the human RefSeq protein database (National Center for Biotechnology Information), using Mascot version 2.4.01 (Matrix Science, London, UK). The retrieved protein was detected by western blot. The primers of LINC01138 and its deletion fragments for *in vitro* transcription were provided in **Supplementary Table 6.** ”

13. - Supplementary Table 3 should include all 3 MS experiments AND include the number of peptides retrieved with the negative control antisense RNA. It would also be helpful to state the approx. MW of the proteins.

Response: Thanks for the Reviewer’s professional comments. We have added the Mass Spectrometry protein identification results for three independent biotinylated LINC01138 RNA pull-down experiments, including the number of peptides retrieved with the negative control antisense RNA, also with the approximate molecular weight of these proteins. Finally, 10 potential interacting proteins were obtained based on peptides number >5 and unique peptides number >5 in the three independent experiments, and absent in the corresponding antisense groups. **The related informations have been provided in the revised Supplementary Table 3, and we have updated the related statements in the revised Results (page 9 in the revised version).**

14. - The experiments in Fig 3b are lacking controls for specificity.

Response: Thanks for the Reviewer’s good point. In the original version, RNA immunoprecipitation (RIP) assays were done in cells transiently transfected with

plasmids containing HA-tagged or FLAG-tagged genes, and confirmed the specific enrichment of OncLn1 in the pCMV-FLAG-IGF2BP1 group, pCMV-FLAG-IGF2BP3 group or pCMV-HA-PRMT5 group, with HA or FLAG antibodies (the original Fig. 3b). According to the reviewer's advice, we further did the RIP assays with commercial antibodies (GAPDH antibody as the negative antibody control), and used PVT1 2036-nt variant as the negative RNA control. **As shown in the revised Figure 3b**, the antibodies of IGF2BP1, IGF2BP3 or PRMT5 could significantly enrich LINC01138, whereas GAPDH antibody and IgG control could not. As expected, there was no obvious enrichment of PVT1 in each group (**the right of the revised Fig. 3b**). **These results demonstrate the specific association between LINC01138 and IGF2BP1/IGF2BP3/PRMT5, and are provided in the revised manuscript (page 9 in the revised version).**

15. - The experiments in Fig 3d have no negative control for specificity (GAPDH or the like)

Response: We appreciate the Reviewer's kind advices. In the revised version, GAPDH was used to be the negative control, and there was no interaction between GAPDH and any of LINC01138 fragments. **The updated results have been provided in the revised Figure 3c.**

16. - Figure 3e,f: The authors state that the C-term of IGF2BP1 is responsible for binding LINC01138. I think the authors mean N-terminal? Still the interpretation is not correct as these domains alone cannot bind the lincRNA.

Response: We appreciate the Reviewer's careful reading. Considering the different length of deletion mutants, we redid the RIP assays in cells transiently transfected with 5 pmol plasmids containing FLAG or HA tagged full length and truncated IGF2BP1, IGF2BP3 or PRMT5, instead of 10 ug plasmids. The updated results showed one representative experiment out of three independent experiments. **As shown in the revised Figure 3d**, the deletion of some K homology RNA binding domains (KH, 158aa-345aa) of IGF2BP1 significantly abolish the association between this protein

and LINC01138, and the deletion of some KH domains (156aa-345aa) of IGF2BP3 also exhibit similar effects (**the revised Figure 3e**), indicating that LINC01138 binds to these regions. **We have updated the results in the revised version (page 10 in the revised version).**

Figure 4:

17. - Figure 4a-c: These experiments should be performed in the CRISPR KO and OE cells. Also, the degree of lincRNA modulation should be displayed.

Response: Thanks for the Reviewer's comments. In the original version, we characterized the molecular regulatory mechanisms between LINC01138 and its binding proteins, in cells which were stably over-expressed LINC01138 via Lenti-virus or transiently transfected with specific LINC01138 siRNAs. **Following the Reviewer's suggestion, these experiments have also been performed in the CRISPR KO and OE cells. The endogenous RNA levels of LINC01138 in the CRISPR KO and OE cells were shown in the revised Supplementary Figure S4a. As shown in the revised Figure 4a, the protein levels of PRMT5 were dramatically reduced in LINC01138-KO-P1 cells and, notably, increased in LINC01138-OE-P1 cells (the revised Supplementary Fig. 8a), whereas LINC01138 had few effect on the protein and mRNA levels of IGF2BP1 and IGF2BP3. Furthermore, with treatment of a protein synthesis inhibitor cycloheximide (CHX), LINC01138 knockout decreased the half-life of the PRMT5 protein, whereas LINC01138 activation increased the half-life of the PRMT5 protein in HCC cells (the revised Figure 4b). Moreover, with treatment of a proteasome inhibitor MG132, the accumulation of endogenous PRMT5 in LINC01138-OE-P1 cells was greater (the revised Figure 4c). These results indicated that LINC01138 might inhibit the proteasome-dependent degradation of PRMT5 in HCC cells, which were consistent with our original results in pWPXL-LINC01138 cells. We have provided the related results in the revised version (page 10 in the revised version)**

18. - Figure 4d: This does not look like ubiquitination? What are the MW of these bands? Analyses of the endogenous PRMT5 should be performed. The CRISPR cells should be employed.

Response: Thanks for the Reviewer's comments. **Accordingly, we evaluated the effects of LINC01138 on PRMT5 protein ubiquitination in the LINC01138 CRISPR KO or OE cells, and have provided the representative images in the revised version. As shown in the revised Figure 4d, the ubiquitination levels of PRMT5 significantly decreased in LINC01138-OE-P1 cells, whereas the ubiquitination levels of PRMT5 increased in LINC01138-KO-P1 cells. These results were consistent with our original results. We have provided the related results in the revised version (page 11 in the revised version). We also labeled the molecular weight of Protein Maker bands in the revised Figures, and the input loading control of HA-tagged PRMT5 have added into the revised figures.**

19. - Figure 4e,f: The CRISPR cells should be employed. Alternatively, 2 independent siRNAs should be used. All panels should display both PRMT5 and CHIP expression.

Response: Thanks for the Reviewer's kind suggestions. There were two independent siRNAs which had been proved to be effective for LINC01138 knockdown (**the revised Supplementary Fig. 4g**). To evaluate the effect of LINC01138 knockdown on the association between PRMT5 and its E3 ligase CHIP, we used the mixture of the two specific siRNAs. The si-LINC01138-mix could significantly knock down the endogenous LINC01138 RNA level (**Fig. R3A**) and decrease the endogenous PRMT5 protein level (**Fig. R3B**). Indeed, LINC01138 knockdown by siRNA mixture notably increased this association between CHIP and PRMT5 in HCC cells (**the revised Fig.4g, h**). **The updated results clearly demonstrate that LINC01138 can block the ubiquitin/proteasome-dependent degradation of PRMT5, and we also added FLAG-tagged-CHIP and HA-tagged PRMT5 into the revised Figures, as the input loading control.**

Figure R3. The effects of LINC01138 specific siRNAs on the endogenous LINC01138 RNA level (A) and the endogenous PRMT5 protein levels (B) .

Figure 5:

20. - Figure 5a: The effect of siRNA-PRMT5 alone should be displayed (moved from suppl.). The observed effects may be independent from the functions of the lincRNA. The interesting (and necessary) experiment would be to assay whether an overexpression of PRMT5 can rescue the effects of knocking out the LINC01138.

Response: Thanks for the Reviewer’s kind suggestions. In the original version, we have confirmed that PRMT5 is a tumor-promoting gene in HCC cells and PRMT5 knockdown by specific siRNAs significantly decreased the proliferation and invasion abilities of SMMC-7721 and SK-Hep1 cells. Furthermore, PRMT5 siRNAs remarkably impaired the colony formation and invasion abilities induced by LINC01138 overexpression (**the revised Fig. 5a and Supplementary Fig. 10a, b**), indicating the contribution of PRMT5 in the LINC01138-induced effects. As kindly suggested by the Reviewer, we then introduced PRMT5 overexpression via lenti-virus into SMMC-7721 cells transfected with LINC01138 siRNAs. Not surprisingly, PRMT5 overexpression could restore the colony formation and invasion abilities which had been reduced by LINC01138 siRNAs (**the revised Fig. 5b and Supplementary Fig. 10c, d**), demonstrating the contribution of PRMT5 in the LINC01138-induced effects. **These results strongly support our findings that LINC01138 is an oncogenic driver**

through activating PRMT5 in HCC cells. We have put the the effect of si-PRMT5 mixture into the revised Figure 5a and 5b, and updated the related results in the revised version (page 11 in the revised version).

21. - Figure 5d: There is barely any methods description for this experiment, so it is unclear how the experiment was performed and the data analysed. According to the supplementary data, two independent siRNA were used for both the lincRNA and PRMT5? At which concentrations and at what time point? What is actually displayed in the figures - the overlap between the two siRNAs? What the same negative control siRNA used in both experiments? Could off-targets of the control siRNA be driving the comparison between the two experiments?

Response: Thanks for the Reviewer's professional comments. **We have added the related informations of RNA seq Assays in the revised Methods, as follows:**

“ 5×10^6 SMMC-7721 cells were transiently transfected with 75nM indicated siRNAs for 48hr, then the total RNA samples were harvested by TRIzol reagent. Before the RNA libraries were constructed, rRNAs in the RNA samples were eliminated using the RiboMinus Eukaryote kit (Qiagen, Valencia, CA, USA). Then, strand-specific RNA-seq libraries were prepared using the NEBNext Ultra Directional RNA Library Prep kit (New England Biolabs, Beverly, MA, USA), according to the manufacturer's instructions. In brief, ribosome-depleted RNA samples were fragmented and prepared for first- and second-strand cDNA synthesis with random hexamer primers. The prepared cDNA fragments were treated with End-It DNA End Repair kit to repair the ends, added an A at the 3'-end by the Klenow fragment, and finally ligated with adaptor sequences. The ligated cDNA products treated with uracil DNA glycosylase to remove the dUTP-labeled second-strand cDNA. The purified libraries were subjected to quality control on a Bioanalyzer 2100 (Agilent, Santa Clara, CA, USA), and sequenced using a HiSeq 3000 (Illumina, San Diego, CA, USA) on a 150 bp paired-end run. For data processed, the raw sequencing reads were aligned to human reference genome (hg19) by using the splice-aware aligner HISAT2. Reads counts for each gene were normalized

into FPKM (Fragments Per Kilobase of transcript per Million mapped reads) values. The cutoff of differential gene expression was $FDR < 0.05$, normalized by the respective si-NC control. The mean FPKM value of the overlap genes of the two subgroups for the two independent siRNAs were used for further analysis.”

The specific siRNAs for LINC01138 were two independent siRNAs, which had been proved to be efficient (**the revised Supplementary Fig. 4g**), so did the specific siRNAs for PRMT5 (**the revised Supplementary Fig. 9a**). The negative control siRNA used in the respective experiment was with the same target sequence “5'-AGUACAGCAAACGAUACGGTTdTdT-3'”, and synthesized freshly before use. **And, the individual result for each siRNA have been shown in the revised Supplementary Figure 11a and b.** The cutoff of differential gene expression was $FDR < 0.05$, then the overlap of deregulated genes by the two specific siRNAs were used to analyze the involved signaling pathways (**the revised Figure 5c**). Though the RNA samples of si-LINC01138 group and si-PRMT5 group for RNA-seq assays were prepared and sequenced separately, there is a strong positive correlation between the gene expression profiling of these two groups ($r=0.9725$, $p<0.0001$; **the revised Fig. 5d**). **We have provided the related results in the revised version (page 12 in the revised version).**

To evaluate whether the off-targets of the negative control siRNA could affect the normalized results, we used the RNA-seq data of another lincRNA, TSLNC8 (Linc00589; Zhang J et al. Hepatology, 2017), as the negative lincRNA control. The siNC for si-TSLNC8 is the same as si-NCs for si-LINC01138 or si-PRMT5. Though the gene expression profiling of the siNC group for si-TSLNC8 is in accord with that of the siNC group for si-LINC01138 ($r=0.9663$, $p<0.0001$; **Figure R4**), there is no obvious correlation between the gene expression profiling of si-TSLNC8 group and that of si-PRMT5 group ($r=0.0678$, $p=0.6930$; **the revised Supplementary Fig. 11e**). These results demonstrate that it is LINC01138 that shares highly similar downstream signalling pathways and gene sets of PRMT5.

Figure R4. The correlation between the si-NC group of TSLNC8 (LINC00589) and the si-NC group of LINC01138, in RNA-seq assays.

22. - What is the actual overlap in affected transcripts between the lincRNA and PRMT5 siRNA experiments?

Response: Thanks for the Reviewer's comments. Our original results showed that LINC01138 and PRMT5 share highly similar downstream signalling pathways and gene sets. And, there were 290 genes whose expression levels significantly altered in both si-LINC01138 group and si-PRMT5 group, for about 63.5% (290/457) of the affected genes in si-LINC01138 group and about 56.9% (290/510) in si-PRMT5 group, strongly supporting our results. **The gene list is shown in the Table R1 in this document.**

23. - Figure 5f,g: The authors should explore the importance of the lincRNA for PRMT5 regulated genes. Can PRMT5 overexpression rescue a knockdown or knockout of the lincRNA?

Response: Thanks for the Reviewer's professional comments. In the original version, we verified the top-scoring genes altered in these two data sets and confirmed that LINC01138 and PRMT5 dramatically affected the similar gene sets that are highly associated with tumorigenesis. Following the reviewer's kind advice, we restored the expression of PRMT5 in si-LINC01138 treated SMMC-7721 cells, and found that

PRMT5 overexpression remarkably reversed the effects of si-LINC01138 on the expression of these detected genes (**the left of the revised Fig. 5e**). On the other hand, PRMT5 knockdown significantly rescued in the expression levels of the genes inhibited by LINC01138 (**the right of the revised Fig. 5e**). **These results clearly demonstrate that LINC01138 regulates the the expression of downstream genes through modulating PRMT5, and the updated results are provided in the revised version(page 12 in the revised version).**

Figure 6:

24. - The authors must demonstrate how an active site-binding small drug can inhibit the association to the lincRNA, which the authors claim prevents binding of PRMT5 to CHIP.

Response: Thanks for the reviewer's comment. **As shown in the revised Figure 3f**, the crucial region of PRMT5 for its interaction to LINC01138 is the S-adenosylmethionine (SAM)-binding domain (350aa-430aa). The inhibitors PJ-68 and HLCL-61, which could specifically inhibit the activity of PRMT5 in leukemia, also bind with the enzymatic active site in "double-E" loop or pyridine ring in the SAM-domain (Jin Yet al., The Journal of clinical investigation, 2016; Tarighat SS et al., Leukemia, 2016), which is exactly the responsible domain for PRMT5 to interact with LINC01138, indicating that these inhibitors might competitively bind to this region thus preventing the association between LINC01138 and PRMT5. To prove this hypothesis, we added another PRMT5 inhibitor, EPZ015666, which is a SAM-uncompetitive, peptide-competitive inhibitor of PRMT5/MEP50 complex formation to inhibit PRMT5 activity (Duncan et al., ACS medicinal chemistry letters, 2016), as the negative inhibitor control. RNA Pull-down assays and RIP assays showed that LINC01138 failed to interact with PRMT5 after treated with PJ-68 or HLCL-61, whereas EPZ015666 has no significant effects on this association in SMMC-7721 cells (**the revised Fig. 6a and 6b**). Moreover, with silencing of endogenous LINC01138, the treatments with PJ-68 or HLCL-61 led to the sharp reduction in HCC cell proliferation (**the revised Fig. 6d**), whereas a slight decrease was observed after EPZ015666 administration with

LINC01138 siRNAs. These results strongly support that an active site-binding feature is crucial for the specific inhibitor to abolish the association between LINC01138 and PRMT5 competitively, such as PJ-68 and HLCL-61, thus leading to an additive therapeutic effect involving of targeting LINC01138-PRMT5 association. **We have added these information in the revised Results (page 13 in the revised version).**

25. - Figure 6d: How do the authors envision this to happen? To this reviewer, an additive effect of siRNA to LINC01138 and the PRMT5 drugs would suggest that they act in different, rather than in the same, pathway.

Response: Thanks for this point. Indeed, the PRMT5 inhibitors and LINC01138 execute their effects in different ways. The inhibitors have been reported to mostly suppress the activity of PRMT5 by binding to the enzymatic active site in SAM-domain or destroying the catalytic complex, while LINC01138 was proved to have the ability to increase the stability of PRMT5 through blocking its ubiquitin/proteasome-dependent degradation in this study. We proposed that some specific PRMT5 inhibitors, such as PJ-68 and HLCL-61 which could abolish the association between LINC01138 and PRMT5 by competitively binding to the SAM domain of PRMT5, could obtain an additive inhibitory effect on HCC cell proliferation with the combination of LINC01138 knockdown, whereas the inhibitor without this domain-specific binding ability could not obtain this advanced therapeutic effect. **We have added this information in the revised Discussion (page 18 in the revised version).**

Reviewer #2 (Remarks to the Author):

In this manuscript, Li et al. characterized an oncogenic Long non-coding RNA *OncLn1*, which is frequently amplified and overexpressed in Hepatocellular carcinoma (HCC). The authors identified *OncLn1* through analysis of RNA-seq data of HCC and adjacent tissues. They also confirmed the tumor-promoting function of *OncLn1* in HCC by either overexpression or knockout/knockdown of *OncLn1* in HCC cells. Subsequently they performed *OncLn1* RNA pull-down assay and mass spectrometry, from which they

found out IGF2BP1, IGF2BP3 and PRMT5 are binding partners of OncLn1 in HCC cells. The authors showed OncLn1 binds to PRMT5 and protects PRMT5 from being degraded by proteasome. In addition, PRMT5 mediates OncLn1-induced HCC cell growth, migration and invasion. While IGF2BP1 and IGF2BP3 act upstream of OncLn1 and enhance RNA stability of OncLn1. In conclusion, the discovery of OncLn1 and OncLn1/PRMT5 axis can be a therapeutic target of HCC.

The authors did many experiments by engaging gain-of-function and loss-of-function approaches to confirm their hypotheses. Most of them are well supported, however, some of the data were not correctly interpreted. Meanwhile, the manuscript should be written carefully and there are several obvious grammar errors in the text. Also, the paper is not readable and hard to understand for non-experts in the field because it lacks of explanations for many treatments/means used in the experiments. Similarly, the authors did not describe their experiments/results clearly either in the text or in the figure legend. The authors should carefully go through the manuscript and rewrite the text to make it more accessible to readers. Here are some specific points need to be addressed/explained as described below in details.

1. There is no Supplementary Fig 2j in the supplementary data so the citations of the figures in the page 7 are not right. The results are not compatible with the figures, which makes it hard to understand. Those highlighted text below should be corrected. HCC cell lines and HCC tissues (Supplementary Fig. 2b-f). In addition, the PhyloCSF codon substitution frequency analysis 17, Coding Potential Assessment Tool (CPAT), Coding Potential Calculator (CPC) <http://cpc.cbi.pku.edu.cn/> and ORF finder software from the National Centre for Biotechnology Information (NCBI) demonstrated that OncLn1 is a bona fide long noncoding RNA (Supplementary Fig. 2g, h). Moreover, OncLn1 is widely expressed in different liver cancer cell lines (Supplementary Fig. 2i) and distributes in both the cytoplasm and nucleus of SMMC-7721, SNU-449 and Huh-7 cells (Supplementary Fig. 2j).

Response: We appreciate the Reviewer's careful reading. **The panels have been**

rearranged in the revised version, and we hope the revised Supplementary Figure 2 and Supplementary Figure 3 would be better for understanding.

2. In Fig 2, what do ctrl-cas9-VP64, OncLn1- (1+3) and OncLn1- (1+4) mean? Although as professionals, we may know the first one is control and the other two are overexpression of OncLn1 in HCC cells. At least it should be explained in the results, figure legends or method part. Similarly, what do OncLn1-sg (1+3) and OncLn1-sg (1+4) mean? The authors should include more detailed information about the experiment design rather than just introduce the experiment name in figure legend.

Response: We appreciate the Reviewer's comments. **The detailed description of "the CRISPR/Cas9 knockout and CRISPR/dead-Cas9 activation technology" has been provided in the revised Method, as kindly suggested by the Reviewer I (Comment 9).** Briefly, we designed the corresponding high-scored sgRNAs for the 4 desired Cas9 cutting sites, which had been numbered as #1, #2, #3 and #4 and shown in **the revised Supplementary Figure 2a**. The combination of paired sgRNAs was used to expect the best efficiency (**the revised Supplementary Fig. 4a**). **In the revised version, the stably cells established via the CRISPR/Cas9 knockout and CRISPR/dead-Cas9 activation technology was nominated as KO and OE, respectively. And, "1#+3#" -sgRNA pair and "1#+4#" -sgRNA pair were identified to be efficive in both knockout system and activation system, nominated as P1 and P2, respectively.** In the original version, OncLn1- (1+3) represented the cells treated with the combination of paired sgRNAs of #1 site and #3 site in the CRISPPR/dCas9 activation system, and we used "LINC01138-OE-P1" instead in the revised version. OncLn1-sg (1+3) represented the cells treated with the combination of paired sgRNAs of #1 site and #3 site in the CRISPPR/Cas9 knockout system, and we used "LINC01138-KO-P1" instead in the revised version. **Following the reviewer's advice, we have also included more detailed information about the experiment design in the revised Figure Legends.**

3. OncLn1 RNA level achieved by overexpression of lentivirus is much higher (almost 100 folds) than endogenous activation, while the colony formation assay showed that

the endogenous activation (Fig2a) indeed had stronger colony formation ability than lentiviral overexpression (Supplementary Fig3f). The colony formation ability of the control cells as well as the parental (Mock) cells are similar according to the figures. Why high OncLn1 level do not achieve stronger colony formation ability?

Response: Thanks for the Reviewer's careful reading. In this study, we established the LINC01138-overexpressed stable cell line with either the CRISPPR/dCas9 activation system or the Lenti-virus system. For the CRISPPR/dCas9 activation system, two sets of paired sgRNAs were used and the activation levels of LINC01138 were about 3-5 folds of the control. For the lenti-virus system, the multiplicity of infection (MOI) was 100 and the overexpression levels of LINC01138 were about 100-200 folds of the control. Each experiment for gain-of-function assays was done more than three times to draw the conclusion, and the representative results from one experiment were shown in the final Figure.

It is a good point whether high LINC01138 level could cause more proliferation ability. To further address this raised issue, we carried out to establish stable Huh7 cell line by the Lenti-virus system with different MOIs, from 10 to 100. **As shown in the Figure R5A**, the RNA levels of LINC01138 in the indicated cells were quite different, and there was no linear dependence between LINC01138 RNA levels and the MOIs. Furthermore, CCK8 assays and colony formation assays showed that LINC01138 overexpression did enhance the proliferation and colony formation abilities of SMMC7721 cells (**Figure R5B, C**), and there was also no linear dependence between LINC01138 RNA levels and their promoting functions on cell growth. Moreover, we performed soft agar assays using the same cells as those used in the other two assays, and found similar oncogenic effects of LINC01138 (**Figure R5B and 5D**). Taken together, we proposed that LINC01138 exhibits the oncogenic effects in a dose-independent way of its over-expressed levels. In our study, the HCC cells with about 3 folds of LINC01138 over-expression levels, driven by the CRISPPR/dCas9 activation system, had gained enough priorities for tumorigenesis.

Figure R5. LINC01138 exhibits the oncogenic effects in an dose-independent way of its over-expressed levels. (A) The relative RNA levels of LINC01138 in different stable lines. (B) CCK8 assays in the indicated cells. (C) The colony formation assays in the indicated cells. (D) The soft agar assays in the indicated cells.

4. In Fig 3c, what are the exact meanings of red boxes. Are they match with the OncLn1 deletions? It is confusing.

Response: Thanks for the Reviewer's careful reading. The red boxes represents the remaining fragments of Linc01138, with the corresponding number label at the corner.

We have updated the image and the labels in the revised Figure S7b, and provided the informations in the related Figure Legend, for better understanding.

readability

5. As is described in the results, 220-630nt fragment of OncLn1 mediates interaction with IGF2BP1 or IGF2BP3. However, in Fig 3d, there is a clear band of IGF2BP1 in #5 (630-2075nt) RNA pull-down complex, while there is no IGF2BP1 or IGF2BP3 in #7 (220-2075nt) RNA pull-down complex. Is that because of the distinguished RNA structures of #5 and #7? Do you have evidence? Otherwise it is not correct to conclude that 220-630nt fragment of OncLn1 mediates its interaction with IGF2BP1 and IGF2BP3.

Response: Thanks for the reviewer's comments. In this study, we constructed a series of Linc01138 deletion mutants based on its secondary structure (<http://www.lncipedia.org/>, **the revised Supplementary Figure S7b**). Actually, there are three paired fragments for full-length Linc01138. They are #2 (1-219 nt) and #7 (220-2075 nt), #3 (1-630 nt) and #5 (631-2075 nt), and #4 (1-1560 nt) and #6 (1561-2075 nt), respectively. We apologize again for the incorrect labels of red boxes in the original version. Indeed, IGF2BP1 and IGF2BP3 showed different patterns of binding abilities in the pull-down assays with Linc01138 fragments. Some fragments (#3 and #4) could interact with both of the two proteins, whereas some ones (#5 and #7) could just interact with either IGF2BP1 or IGF2BP3. The fragments, which could interact with both IGF2BP1 and IGF2BP3, share the 220-630 nt sequence of Linc01138. **We have updated the image and the labels in the revised Figure S7b, and modified the related statements in the revised Results (page 10 in the revised version),** as "The 220-1560nt fragment of LINC01138 mediates the interaction with IGF2BP1 or IGF2BP3, and the fragments, which could interact with both IGF2BP1 and IGF2BP3, share the 220-630 nt sequence of Linc01138,..."

6. In page 9, text showing below is confusing.

Next, RIP assay results showed that the RNA recognition motif (RRM) and K homology

RNA binding domain (KH) at the C-terminus of IGF2BP1 and IGF2BP3 (Fig. 3e, f), mediated their interactions with OncLn1, whereas the S-adenosylmethionine (SAM)-binding domain at the N-terminus of PRMT5 physically associates with OncLn1 in the HCC cells (Fig. 3g).

What does it mean? It is obvious that in Fig 3e and 3f, RRM alone is not efficient to pull down OncLn1. SAM domain (350-440 aa) as showing in Fig 3g is not in the N-terminal of PRMT5 (637aa). There is no significance analysis showing in the figure, readers will unable to tell what exactly the authors want to say. Is 5 folds significant or 10 folds significant compare with IgG control? It will be different interpretations regarding different definition of significance.

Response: Thanks for the reviewer's comments. We apologize for the incorrect interpretations in the original version. In this study, RIP assay results showed that the deletion of some K homology RNA binding domains (KH, 158aa-345aa) of IGF2BP1 significantly abolish the association between this protein and LINC01138 (**the revised Figure 3d**), and the deletion of some KH domains (156aa-345aa) of IGF2BP3 also exhibit similar effects (**the revised Figure 3e**), indicating that LINC01138 binds to these regions. **And as shown in the revised Figure 3f**, the crucial region of PRMT5 for its interaction to LINC01138 is the S-adenosylmethionine (SAM)-binding domain (350aa-430aa), and the LINC01138-binding abilities of the mutants without this domain were definitely dampened. **We have updated the images in the revised Figure 3d-3f and the related statements in the revised version (page 10 in the revised version).**

7. Many labels of western blot figures are confusing. For example, in Fig 4e and 4f, what does the IgG do here? The figure labels showed that these experiments used FLAG or HA antibodies for immunoprecipitation, so IgG was also used for immunoprecipitation? Alternatively, IgG means a plasmid used here as a control of pCMV-FLAG-CHIP/ pCMV-HA-PRMT? In addition, there is no explanation about it in figure legend. There are lots of mixed use of plasmid names and protein names for western blot labels. The authors should check carefully and try to make it clear and

correct.

Response: Thanks for the reviewer's professional comments. **We have provided the updated images with rearranged labels in the revised version for better readability. And, we have also included more detailed information about the experiment design in the revised Figure Legends.**

Reviewer #3 (Remarks to the Author):

Through analyzing TCGA database, Li et al discovered that LINC01138 (they named it lincRNA OncLn1 in this paper) was significantly increased in HCC tissues which was associated with the poor outcome of HCC patients. They further studied the biological functions and mechanisms of OncLn1 in HCC tumorigenesis. The authors found that OncLn1 exerted oncogenic functions in HCC tumorigenesis through promoting HCC cell growth and metastasis. RNA pull-down assay showed that IGF2BP1, IGF2BP3 and PRMT5 physically associate with OncLn1. The authors further showed that OncLn1 increased PRMT5 stability through blocking its ubiquitination.

This study is potentially interesting, however LINC01138, as a novel oncogenic long intergenic non-coding RNA (lincRNA), promoted proliferation with anti-apoptosis activity in prostate cancer cells, also as a potential biomarkers, has just been reported recently from the same city, which thus spoils the novelty of the current study.

Some of their data were over-interpreted, and several key scientific questions have not been addressed adequately. The conclusion of oncogenic IGF2BP1/IGF2BP3-OncLn1-PRMT5 axis in HCC still lacks solid experimental evidence.

Comments:

1. The authors found that OncLn1 upregulates PRMT5 expression through inferring it binding with E3 ligase CHIP and further blocking its degradation. As substrate specificity of proteasome-dependent degradation, it seems that E3 ligase CHIP may be a key regulator in OncLn1-PRMT5 axis-mediated oncogenic behavior. However, the authors didn't show the expression levels of CHIP in HCC tumor samples, and its regulation by OncLn1.

Response: Thanks for the reviewer's comments. We analyzed the protein levels of CHIP in a 239-patient cohort (Wang Z et al., Hepatology, 2015), using immunohistochemical (IHC) staining. There was no significant difference between the IHC scores of HCC tissues and those of the corresponding non-tumor tissues ($p=0.2641$, **Fig. R6A**), and the survival analysis showed no significant differences in the patients with high or low CHIP expression ($p=0.2289$, **Fig. R6B**). However, CHIP-high patients correlated with small tumor size ($<5\text{cm}$, $p=0.035$, **Table R2 in this document**), indicating that patients with low CHIP expression were more likely to exhibit aggressive clinicopathologic features. In our study, LINC01138 itself has no obvious effect on the protein levels of CHIP (**the revised Fig. 4f and 4h**). **We demonstrated that LINC01138 interacts with PRMT5, thus inferring its binding with E3 ligase CHIP and further blocking the function of CHIP on PRMT5 degradation. Though the protein levels of CHIP is not frequently deregulated in HCC tissues, LINC01138-induced blockage of the interaction between CHIP and its substrate, PRMT5 which have been proved to have oncogenic effects on HCC cells, would be a definite advantage for HCC progression.**

Figure R6. The protein levels and the clinical significance of CHIP in HCC. (A) IHC staining scores for CHIP. The IHC scores (0, 1, 2 and 3) are based on the intensity of staining. Values are expressed as mean \pm SD. The statistical analysis was performed using Wilcoxon test. (B) Kaplan-Meier analysis of the correlation between CHIP protein levels and the overall survival of 239 HCC patients. The statistical analysis was performed using Log-rank test. Patients were stratified for the analysis by the IHC score =3 (high), or <3 (low).

2. As no effects were observed on expression levels of IGF2BP1 and IGF2BP3 by OncLn1 in HCC cells, the authors showed that IGF2BP1 and IGF2BP3 could be important upstream effectors of OncLn1 in HCC cells through enhancing OncLn1 stability. However, the authors didn't find key regulators mediating stabilization of OncLn1 by IGF2BP1 and IGF2BP3, as IGF2BP1 and IGF2BP3 should be adaptor proteins.

Response: Thanks for the reviewer's comments. In our study, we found three candidate binding proteins IGF2BP1, IGF2BP3 and PRMT5 for LINC01138. Among them, only the protein levels of PRMT5 were remarkably affected by LINC01138 overexpression or knockdown (the revised Fig. 4a and Supplementary Fig. 8e), whereas the expression levels of IGF2BP1 and IGF2BP3 (both mRNA and protein) were not altered under the same conditions (the revised Fig. 4a and Supplementary Fig. 8a, b). Besides studying the molecular mechanisms through which LINC01138 regulates

PRMT5 protein, we evaluated the biological consequences of the interaction between LINC01138 and IGF2BP1/IGF2BP3. It turned out that IGF2BP1 and IGF2BP3 could regulate the RNA levels of LINC01138 (**the revised Fig. 7a**), and our experiments revealed the strong impacts of IGF2BP1 and IGF2BP3 on LINC01138 RNA stability (**the revised Fig. 7b**), indicating that these two proteins are the regulators of LINC01138. Moreover, LINC01138 siRNAs could significantly inhibit the promoting effects induced by IGF2BP1 or IGF2BP3 overexpression (**the revised Fig.7c and 7d**), indicating LINC01138 is involved in the oncogenic effects of these two proteins on HCC cells.

IGF2BPs, including IGF2BP1, IGF2BP2, and IGF2BP3, are well-known RNA binding proteins which were shown to regulate translation, localization, or stability of their target RNAs (Doyle GA et al., *Nucleic Acids Res*, 1998; Nielsen J et al., *Mol Cell Biol*, 1999; Houseley J et al., *Cell* 2009). Specifically, IGF2BP1 could stabilize MYC, MDR1, and PTEN mRNAs (Lemm I et al., *Mol Cell Biol*, 2002; Sparanese D et al., *Nucleic Acids Res*, 2007; Stohr N et al., *Genes Dev*, 2012). IGF2BPs could also regulate the stability of lncRNAs, and all the three IGF2BPs were reported that could modulate the stability of HULC in HCC (Hänmerle M et al., *Hepatology*, 2013). In our study, IGF2BP2 could not interact with LINC01138 (**Fig R7A**), and had few effects on the RNA levels of LINC01138 (**Fig R7B and R7C**), which is quite dissimilar to the effects of IGF2BP1 and IGF2BP3 on LINC01138. **Our results demonstrated that the binding of LINC01138 to IGF2BP1 and IGF2BP3 is specific for the contribution on the stability of this lincRNA, further for the biological consequences, such as modulating the protein levels of PRMT5 (the revised Fig.7e). Though there might be other regulators of LINC01138, our study revealed that IGF2BP1 and IGF2BP3 have strong impacts on LINC01138 RNA stability, and act as the specific regulators of this lincRNA.**

Figure R7. The effect of IGF2BP2 on LINC01138 in HCC cells. (A) Immunoblotting for the specific associations of IGF2BP2 with biotinylated-LINC01138 from three independent streptavidin RNA pull-down assays. (B) The RNA levels of LINC01138 in SMMC-7721 cells treated with siRNAs for IGF2BP2. (C) The efficiency of siRNA for IGF2BP2. Values are expressed as mean \pm SEM.

Table R1. Genes significantly affected by both linc01138 knockdown and PRMT5 knockdown in RNA-seq analysis

Gene ID	Signal Intensity (FPKM)					Signal Intensity (FPKM)				
	Linc01138_siNC	linc01138_si1	linc01138_si2	LogFC(si1/NC)	LogFC(si2/NC)	PRMT5_siNC	PRMT5_si1	PRMT5_si2	LogFC(si1/NC)	LogFC(si2/NC)
NUPR1	6.17051	116.86374	70.6837	18.939073	11.455082	26.23491	97.59974	98.47109	3.7202239	3.7534373
ECM2	3.11221	27.92004	31.19426	8.9711298	10.023186	35.95664	55.89221	53.36009	1.5544336	1.4840121
CYP1A1	15.37078	119.62539	85.59124	7.7826493	5.5684383	0.45868	1.42824	1.30102	3.1138048	2.8364437
S100A14	1.06015	7.39724	5.06004	6.9775409	4.7729472	30.86584	139.73756	121.67863	4.527256	3.9421778
NPW	1.1348	7.35251	4.5136	6.4791241	3.977441	24.1238	53.59732	59.153	2.2217611	2.4520598
ETV5	3.73336	21.97945	19.67531	5.8873106	5.2701347	30.08534	144.55203	40.61799	4.8047331	1.3500924
RND3	10.9239	56.88495	40.7525	5.2073847	3.7305816	217.4266	479.12992	284.80344	2.2036394	1.3098829
TSACC	1.21418	6.05143	7.72696	4.9839645	6.3639329	31.98368	71.79917	114.75175	2.2448689	3.587822
PSAT1	199.8541	976.02112	889.90677	4.883667	4.452781	1524.653	2491.5405	2483.9077	1.6341681	1.6291619
RSG1	1.75251	8.2977	7.66757	4.7347519	4.3751933	58.3126	83.97325	82.65733	1.4400533	1.4174866
ANKRD1	3.42044	16.02451	17.41396	4.6849265	5.0911462	30.98194	236.69417	84.64947	7.6397466	2.7322198
PHGDH	18.30125	85.45195	61.10732	4.6691865	3.3389697	51.19121	121.80401	127.8527	2.3793931	2.4975518
PHLDA1	5.87227	25.25341	18.5519	4.3004511	3.1592383	60.94404	151.48684	159.05704	2.4856711	2.6098867
CYP3A5	2.6147	11.02671	9.59986	4.2171989	3.6714958	4.25248	8.84391	7.84016	2.0797064	1.8436677

EGR1	3.88282	16.34156	12.35501	4.2086834	3.1819683	2.37437	4.23256	4.61807	1.7826034	1.9449665
SERPINE2	6.5475	26.63725	18.89589	4.0683085	2.8859702	59.25021	422.22852	116.72256	7.1261945	1.969994
ULBP1	9.04839	36.7217	39.41714	4.0583684	4.3562601	50.97558	151.92234	173.64944	2.9802964	3.4065221
MKX	1.31886	5.25852	6.55714	3.9871707	4.9718242	20.26568	34.22439	41.95547	1.6887857	2.070272
PLAU	1.83568	7.31915	6.93625	3.9871601	3.7785725	24.81374	37.80611	32.96527	1.5235958	1.3285087
OR51B5	3.80997	14.55806	20.20534	3.8210432	5.3032806	36.38839	62.77376	89.22892	1.7251041	2.4521261
GPT2	20.8803	75.50144	67.62684	3.6159174	3.2387868	150.7496	447.08817	455.02616	2.9657655	3.0184223
GNG3	1.53493	5.35503	6.10513	3.488778	3.9774648	21.98981	10.11592	14.48645	0.4600276	0.6587801
THBS1	4.48364	15.64242	14.56402	3.488777	3.2482581	52.3194	95.21437	99.4904	1.8198674	1.9015967
ABCG2	1.95476	6.81972	8.88568	3.4887761	4.5456629	13.93765	50.61091	21.96275	3.631237	1.5757857
CCDC112	13.58339	46.77392	39.94748	3.4434644	2.9409065	44.40799	69.74429	62.87564	1.5705347	1.4158632
STC2	54.78869	187.84217	173.80847	3.4284844	3.1723421	339.4699	836.0963	810.51874	2.4629467	2.3876011
SPX	2.30632	7.81634	5.13704	3.389096	2.2273752	15.34803	33.22727	30.80247	2.1649208	2.0069331
CCDC7	1.16373	3.79464	5.29955	3.2607564	4.5539343	0.54499	3.39895	4.32662	6.2367199	7.938898
SENP7	4.65663	14.98003	11.95483	3.2169251	2.5672708	86.27571	339.1843	225.86305	3.9313997	2.6179217
PI3	7.03888	22.21829	16.7981	3.1565093	2.3864734	54.21539	139.16844	135.95704	2.5669545	2.5077204
SNAPC1	6.11661	19.05312	23.11214	3.1149804	3.7785865	69.25444	102.45799	97.549	1.4794429	1.4085595

RNF224	9.00739	27.38443	22.21243	3.0402181	2.4660229	61.19089	373.79637	248.46112	6.1086931	4.0604266
CD22	3.47785	10.16897	11.0664	2.9239243	3.1819659	2.52978	5.04145	4.45653	1.9928413	1.7616275
WFIKKN1	1.38767	4.01134	4.85707	2.8907017	3.5001621	31.16685	41.15426	47.50287	1.3204498	1.5241473
CBS	21.19973	60.43281	47.59802	2.8506406	2.2452182	28.21894	52.62276	40.41315	1.8648029	1.4321286
ELOVL6	6.12248	17.2914	16.38219	2.8242477	2.6757441	111.4593	207.81295	232.80742	1.8644735	2.088721
RAD9B	1.70398	4.67091	6.09975	2.7411765	3.5797075	24.4116	81.67832	103.02547	3.3458815	4.2203489
ZNF559	1.43109	3.85155	7.0582	2.6913402	4.9320448	31.8775	130.24594	142.33713	4.0858267	4.4651284
PLCD1	1.56909	4.22295	4.24387	2.691337	2.7046696	19.57862	66.67163	101.7029	3.4053284	5.1945898
SIPA1L2	15.76877	42.20953	42.28287	2.6767801	2.6814311	115.2871	277.33629	256.32395	2.4056136	2.2233527
ZNF675	4.58919	12.13229	14.60264	2.643667	3.1819646	62.42624	83.17345	89.36202	1.3323476	1.4314817
ZNF267	5.78094	15.15298	16.01025	2.6211966	2.769489	69.36226	143.22437	177.68568	2.0648746	2.5617055
ZNF501	1.38497	3.62389	3.85607	2.6165838	2.7842264	20.00146	118.89226	92.7248	5.9441791	4.6359016
LYPD6	1.07595	2.8153	4.06556	2.6165714	3.7785771	23.37003	42.54596	34.83598	1.8205351	1.4906262
ARRDC4	38.13552	99.29311	98.07049	2.603691	2.5716311	279.0944	540.57787	378.36087	1.9368994	1.3556732
SLC27A6	1.93789	5.02236	5.85799	2.5916641	3.0228702	38.86776	64.82256	62.15032	1.667772	1.5990199
ZNF805	4.25894	10.87853	9.31686	2.5542811	2.1876007	39.61028	67.53959	74.61253	1.7051026	1.8836658
TRIM23	7.6041	19.37036	18.81909	2.5473573	2.4748609	130.5434	260.9536	283.91902	1.9989784	2.1749

ENDOD1	3.75497	9.50127	7.81227	2.5303185	2.0805146	310.1536	654.1375	498.41586	2.1090755	1.6069965
OR51B2	1.29924	3.23769	6.20122	2.4919876	4.7729596	13.80984	19.26588	19.75551	1.3950835	1.4305387
NIPAL1	1.58446	3.94845	4.1414	2.4919847	2.6137612	29.39415	48.48234	82.15284	1.6493874	2.7948704
SNAPC5	15.10829	37.64958	41.63557	2.4919816	2.7558096	205.2237	327.60039	328.11396	1.5963086	1.5988111
HIST1H1E	1.7246	4.29767	8.23142	2.4919807	4.7729445	7.17301	15.68296	14.468	2.1863848	2.0170054
PVR	78.09707	193.84951	182.39814	2.4821611	2.3355312	517.5488	803.73716	854.71698	1.5529688	1.6514712
SAMD9	18.77	46.47907	50.29514	2.4762424	2.6795493	253.7462	412.14112	429.50517	1.6242257	1.6926564
TRIML2	8.43358	20.79509	24.71676	2.4657488	2.9307554	70.97488	152.4747	163.97678	2.1482911	2.3103495
PYROXD1	12.30483	29.68775	32.7021	2.4126908	2.6576637	146.4276	240.93344	248.53252	1.645409	1.6973055
ZNF546	2.42923	5.81146	5.7973	2.3923054	2.3864764	35.69986	261.15196	262.17239	7.3152096	7.3437932
SRPK3	2.85373	6.82699	6.58335	2.3923041	2.3069281	17.804	54.20173	69.80663	3.0443569	3.9208397
SGOL2	17.53524	41.45661	39.88022	2.3641883	2.2742899	259.9410	589.29368	582.09983	2.2670278	2.2393529
RTEL1	1.50447	3.53488	3.07747	2.3495849	2.0455509	8.44259	79.32864	101.1215	9.3962445	11.977545
TLR3	1.20804	2.80972	2.72279	2.3258501	2.2538906	12.28216	44.13591	31.33218	3.5934974	2.5510317
ZNF563	1.42657	3.31798	3.02619	2.3258445	2.1213049	19.77805	28.98867	25.93008	1.4656991	1.3110534
ZDHHC1	1.24546	2.89674	2.642	2.3258394	2.1213046	14.77299	22.99621	21.33209	1.5566388	1.4439927
SPOCK1	5.26263	12.13079	13.43128	2.3050813	2.5521992	41.3449	119.58465	77.14609	2.8923676	1.8659155

DAAM1	7.58296	17.44785	16.58849	2.3009287	2.1876009	0.01	1.40044	1.66429	140.044	166.429
ELFN1	1.76048	4.03612	3.64117	2.2926247	2.0682825	71.95423	117.09047	94.48853	1.627291	1.3131755
RUNX2	1.15696	2.636	2.76106	2.2783847	2.3864784	8.63117	25.18921	33.97833	2.9184004	3.9367004
ZC3H15	76.67984	173.16382	174.03426	2.2582705	2.2696221	832.8594	1117.7786	1175.6537	1.3420976	1.4115872
SPRED1	7.86718	17.5627	18.12292	2.2324009	2.3036107	67.63941	123.13992	121.38193	1.8205351	1.7945445
ZNF674	6.11103	13.59694	13.02126	2.2249833	2.1307799	45.89211	171.34831	130.12856	3.7337205	2.8355323
NR5A2	2.43148	5.36671	6.35529	2.2071783	2.6137538	20.54718	31.19065	29.86866	1.5180015	1.4536623
PIBF1	28.04296	61.60698	73.08782	2.1968786	2.6062805	319.7289	424.95557	425.41406	1.329112	1.330546
TERT	7.91874	17.39458	19.83109	2.1966348	2.5043239	23.99031	65.39765	70.96637	2.7260027	2.9581264
ZNF239	13.48955	29.58182	33.05091	2.1929434	2.4501121	124.4309	184.02793	162.95957	1.4789567	1.309639
TBX20	2.1661	4.67816	8.61556	2.1597156	3.9774526	23.69111	46.9156	37.9838	1.980304	1.6032934
NRIP1	13.81092	29.3178	27.94088	2.1227985	2.0231006	122.0560	239.20631	286.79509	1.9598071	2.3496999
CARS	60.29226	127.25636	132.83427	2.1106583	2.2031728	406.4901	551.34708	554.40641	1.3563601	1.3638863
RSL24D1	143.97456	301.82533	294.20025	2.0963796	2.0434183	1281.232	2097.1274	2543.05	1.6368049	1.9848468
HDAC10	1.36721	2.86193	2.82777	2.0932629	2.0682777	4.80197	19.45871	26.38495	4.0522348	5.4946095
PAK1IP1	48.88761	101.95986	110.08768	2.0855971	2.2518524	535.6221	1507.6741	1271.2518	2.8148091	2.3734116
RNF146	12.34947	25.58143	25.48073	2.0714597	2.0633056	143.5703	772.11719	614.28117	5.3779718	4.2786081

SYCP2	11.09681	22.8529	25.64947	2.0594117	2.3114273	134.7857	243.66589	285.65757	1.8078022	2.1193463
C9orf85	9.52813	19.53785	22.52207	2.050544	2.363745	39.28047	55.00409	52.85768	1.400291	1.3456478
NBN	32.10792	65.32516	65.60336	2.0345497	2.0432143	315.8717	795.4184	804.88529	2.5181688	2.5481395
CCDC82	22.09512	44.91789	54.88795	2.0329326	2.4841662	0.1	1.89111	0.60181	18.9111	6.0181
PLEKHN1	6.21233	12.62299	18.627	2.0319252	2.9983919	65.14841	114.95301	143.30292	1.7644791	2.199638
MPHOSPH10	37.37248	75.22718	81.35277	2.0129031	2.1768095	421.8723	622.915	656.46421	1.4765487	1.5560733
TG	0.12161	0.24245	0.19349	1.9936683	1.5910698	0.33721	1.40262	0.51012	4.1594852	1.5127665
PCDH9	0.1962	0.39115	0.62432	1.993629	3.1820591	0.72539	0.32327	1.13162	0.4456499	1.560016
JAKMIP2	0.21786	0.43433	0.34662	1.9936198	1.5910218	0.50342	1.07687	0.79962	2.1391085	1.5883755
TMEM92	0.43685	0.87091	0.69503	1.9936134	1.5910038	3.63395	5.03838	1.83243	1.3864748	0.5042529
CA8	0.2341	0.4667	0.7449	1.9935925	3.1819735	0.10819	0.77142	0.24549	7.1302338	2.2690637
IL7R	2.90421	5.7898	7.46397	1.9935886	2.5700518	37.58003	154.9614	69.69512	4.1235039	1.8545786
IFT74	1.44214	2.87503	2.86803	1.9935859	1.988732	8.16422	11.28646	11.53129	1.3824297	1.4124178
C6orf99	3.15942	6.29857	5.02658	1.9935843	1.5909819	35.77203	54.65785	23.19184	1.5279494	0.6483233
TAS2R19	1.35111	2.69355	2.14959	1.9935831	1.5909807	9.36595	20.03498	12.75155	2.1391295	1.3614796
FOLR2	0.95676	1.90738	3.04437	1.9935825	3.1819579	10.1695	18.91642	7.02311	1.8601131	0.6906052
LTK	0.38791	0.77333	0.61716	1.993581	1.5909876	1.97195	3.83477	3.25426	1.9446588	1.6502751

TLE6	1.11364	2.22013	2.06707	1.9935796	1.8561384	13.55251	7.951	21.21528	0.586681	1.5654133
ADRB2	0.66331	1.32236	2.11062	1.9935777	3.1819511	8.27655	10.92876	15.30269	1.3204487	1.8489214
SLC8A3	0.16033	0.31963	0.25508	1.9935758	1.5909686	1.03731	0.26416	0.58845	0.2546587	0.5672846
CCDC102B	0.46332	0.92366	1.28997	1.9935682	2.784188	2.99762	4.19852	4.12978	1.4006178	1.3776863
TCEB3B	0.44446	0.88606	0.70712	1.9935652	1.5909643	1.43779	2.19688	3.0295	1.5279561	2.1070532
UNC5CL	0.45339	0.90386	0.72133	1.9935596	1.5909702	1.88574	2.98803	1.18861	1.5845398	0.6303149
NOV	36.97276	72.95108	68.49255	1.9731034	1.8525139	185.6105	431.42534	278.57014	2.3243581	1.5008316
ELFN2	11.25349	22.12748	14.83834	1.9662771	1.3185545	71.95423	117.09047	94.48853	1.627291	1.3131755
VEPH1	10.32813	20.28115	18.23937	1.9636807	1.7659896	58.03975	76.23522	76.68049	1.3135001	1.321172
ZNF449	2.03199	3.92435	4.64723	1.9312841	2.2870339	29.58024	48.96353	41.4182	1.6552783	1.4001982
BCL6B	1.8575	3.5488	2.46271	1.9105249	1.3258197	10.73026	15.81234	15.09599	1.4736213	1.4068615
TMEM253	3.45184	6.53746	4.94263	1.8939059	1.4318827	34.13772	10.23715	17.73685	0.299878	0.5195675
DLC1	0.36962	0.67547	0.8821	1.8274715	2.3865051	1.87899	0.609	3.29465	0.3241103	1.7534154
RASEF	0.61083	1.11627	1.45774	1.8274643	2.3864905	0.94096	2.34831	0.42703	2.4956534	0.4538238
AKR1B10	4.65761	8.51156	6.79266	1.8274523	1.4584003	38.38537	26.85882	57.79628	0.699715	1.5056851
IL12A	1.9909	3.63827	3.16748	1.8274499	1.590979	4.60034	10.93412	7.65509	2.3768069	1.664027
TSPAN8	10.21136	18.4822	25.22419	1.8099646	2.4702087	172.3692	250.59496	282.35637	1.4538267	1.6380905

NT5E	13.48081	24.01967	24.12871	1.7817676	1.7898561	97.34341	284.30349	147.72789	2.9206239	1.5175952
PAPSS2	54.27084	95.40704	84.51242	1.7579798	1.5572344	487.6265	1042.6625	787.442	2.1382397	1.6148463
HEATR4	1.28659	2.24431	0.7676	1.7443863	0.5966159	5.35121	3.1797	3.54159	0.5942021	0.6618298
NBPF24	0.37627	0.65636	0.59864	1.7443857	1.5909852	2.69525	4.02964	0.98643	1.4950895	0.3659883
TMEM156	5.2842	9.21768	13.87165	1.7443851	2.6251183	36.87451	70.52124	56.79796	1.9124658	1.5403041
MADCAM1	1.10923	1.93492	3.52952	1.7443812	3.181955	7.1766	3.65516	4.07116	0.5093164	0.5672826
CTSC	4.62294	7.97558	7.8265	1.7252182	1.6929703	28.75965	42.96676	38.59628	1.4939945	1.3420288
PSD3	9.61286	16.54674	14.0902	1.7213129	1.4657657	60.1788	97.96283	81.99749	1.6278628	1.3625644
RBMS3	2.48478	4.17146	3.95324	1.6788046	1.5909819	20.9717	28.87318	28.31939	1.3767687	1.3503622
PCLO	0.17759	0.29503	0.23545	1.6612985	1.3258066	0.10943	1.07285	0.49661	9.8039843	4.5381522
UNC13A	4.14263	6.84291	5.46099	1.6518275	1.3182423	20.12918	31.78715	13.1566	1.5791577	0.6536083
DEPTOR	6.71226	11.06544	3.28587	1.6485416	0.4895326	90.43466	167.58981	126.15697	1.8531591	1.3950068
KRT15	1.91514	3.14984	2.8946	1.6447048	1.51143	16.99306	32.50074	10.94527	1.912589	0.6441024
EEPD1	15.362	25.19639	23.55192	1.6401764	1.5331285	89.58078	186.8388	124.18835	2.0857019	1.3863281
PLD6	3.70183	6.06207	5.04818	1.6375874	1.3636985	9.77571	23.52545	5.82287	2.4065209	0.5956468
TRAF1	1.46948	2.4064	2.17092	1.6375861	1.4773389	14.35812	55.34032	8.14512	3.8542873	0.5672832
ANKRD18A	2.04503	3.33568	4.14096	1.6311154	2.0248896	10.82551	16.23451	17.05868	1.4996531	1.5757853

MAL2	46.56067	75.91007	110.48806	1.6303475	2.3729912	446.2124	1531.6821	701.76735	3.4326299	1.5727204
SOX7	1.46318	2.37005	2.61889	1.6197939	1.7898618	14.36906	30.73726	19.75496	2.1391281	1.3748262
PIWIL2	4.94349	7.97808	10.67394	1.6138558	2.1591912	40.46951	67.87483	58.87529	1.6771844	1.4548061
MVB12B	1.67758	2.67553	2.53556	1.5948748	1.5114391	7.98532	2.76402	5.01371	0.3461377	0.6278659
MUC5B	0.36371	0.58007	0.1736	1.5948695	0.4773033	4.03404	1.91763	2.21216	0.4753622	0.5483733
SPOCD1	2.10189	3.3332	3.19207	1.5858109	1.5186665	10.50837	46.59464	17.03206	4.4340502	1.6208089
RAP1B	40.61673	64.32316	61.75273	1.5836617	1.5203767	748.9309	1451.6662	1139.2912	1.9383178	1.5212233
MYPN	3.01204	4.70373	6.86868	1.5616426	2.280408	18.55964	33.74625	29.58527	1.8182599	1.5940649
ZNF680	6.54366	10.14638	11.76041	1.5505665	1.797222	71.34555	110.21031	96.83048	1.5447398	1.3572042
IGDCC4	0.17437	0.26072	0.41613	1.4952113	2.386477	0.40291	1.72377	1.46283	4.2783004	3.6306619
FLNC	0.72942	1.09063	1.04445	1.4952017	1.4318911	0.47193	2.64398	0.76491	5.6024834	1.6208124
CLEC18B	0.5187	0.77556	0.82524	1.4951995	1.5909774	1.91769	0.85462	0.54394	0.4456508	0.2836433
ERG	0.33845	0.50605	0.80771	1.4951987	2.3864973	2.26797	1.3941	3.10553	0.6146907	1.3692994
LRRC34	0.27908	0.41728	1.33203	1.4951985	4.7729325	1.28972	2.29907	0.58531	1.7826117	0.4538272
GDF9	0.43629	0.65234	1.73533	1.4951981	3.9774691	8.66992	3.5942	5.49019	0.4145598	0.6332458
HLA-G	0.7679	1.14816	3.0543	1.4951947	3.977471	3.90363	7.59124	5.23419	1.9446618	1.340852
TMEM220	0.77184	1.15405	1.84198	1.4951933	2.3864791	6.06384	8.9019	8.09391	1.4680302	1.3347829

GRM1	0.18617	0.27836	0.29619	1.4951926	1.5909652	1.54863	3.37406	2.34269	2.1787386	1.51275
SKAP1	1.53377	2.29328	0.4067	1.4951916	0.2651636	2.59898	5.05414	6.96969	1.9446629	2.6817021
SLC16A12	0.27971	0.41822	0.89004	1.4951914	3.1820099	5.68769	10.59977	10.85286	1.8636336	1.9081314
HIST2H3D	3.29394	4.92507	5.24061	1.4951912	1.5909853	7.61126	27.13578	20.72514	3.5652152	2.7229578
REC8	2.8197	4.21599	1.96267	1.495191	0.6960563	6.51544	8.71086	11.82752	1.3369565	1.8153064
AIM1	1.69036	2.52741	2.28594	1.4951904	1.3523391	11.24897	16.98891	15.24432	1.5102636	1.3551747
ZCCHC12	3.64091	5.44385	2.41359	1.4951894	0.6629084	0.8413	3.99921	0.31817	4.7536075	0.3781885
NLRP9	0.37616	0.56243	1.19694	1.4951882	3.181997	2.2599	0.61977	1.18339	0.2742466	0.5236471
FSIP1	1.37629	2.05781	3.28448	1.4951863	2.3864738	15.68884	9.82626	20.68651	0.6263216	1.3185494
CKLF-										
CMTM1	2.15919	3.22839	0.85881	1.4951857	0.3977464	8.98056	5.33627	5.09452	0.5942024	0.5672831
RAB41	1.0073	1.5061	1.6026	1.4951851	1.5909858	7.44816	9.95786	17.42901	1.3369557	2.3400424
TEKT3	0.5191	0.77615	0.82588	1.495184	1.5909844	1.43937	3.42111	4.89918	2.3768107	3.4036975
PRSS16	0.76776	1.14794	1.01791	1.4951808	1.325818	6.62309	12.64968	4.2939	1.9099363	0.6483228
RHBDL3	0.25052	0.37457	0.59786	1.49517	2.3864761	2.19972	1.23828	0.78812	0.5629262	0.358282
FGF1	0.26803	0.40075	0.42643	1.4951685	1.5909786	0.74319	3.97447	2.52961	5.3478518	3.4037191
PCSK1	13.52025	19.83391	24.35149	1.4669781	1.8011124	110.8172	236.21164	171.33973	2.1315431	1.5461474

PPT1	272.07335	392.91431	363.76219	1.4441485	1.3370004	1888.298	3006.9364	2542.2184	1.5924053	1.3463012
DCBLD2	5.76396	8.31897	10.88981	1.4432734	1.8892931	50.33356	155.51009	65.85848	3.0895905	1.3084407
ACSS1	2.27959	3.27476	3.09344	1.4365566	1.357016	19.83029	10.82582	13.35864	0.5459234	0.6736482
BMPR1B	2.57378	3.6345	4.26547	1.4121254	1.6572784	12.78648	23.32335	18.10577	1.8240634	1.4160089
VCAN	0.5477	0.77341	1.30706	1.4121052	2.3864524	0.50622	0.90239	0.19145	1.7826044	0.3781953
FAM35A	57.446	80.0074	81.4944	1.392741	1.4186262	281.7027	380.17337	387.04637	1.3495551	1.3739531
FAM175A	6.49793	9.04559	9.26864	1.3920726	1.4263989	43.07656	60.17567	61.56142	1.396947	1.4291164
LRIG3	6.89196	9.5033	14.25449	1.3788966	2.0682781	46.81991	63.21121	64.44294	1.3500925	1.3764003
CYP3A7	1.53363	2.10198	2.13498	1.3705913	1.3921089	4.25248	8.84391	7.84016	2.0797064	1.8436677
AC006116.20	2.44481	3.35083	3.40344	1.3705891	1.3921082	11.5808	5.03513	6.72982	0.4347826	0.5811187
SWT1	3.33944	4.49379	4.51605	1.3456717	1.3523375	33.79783	53.3706	47.6259	1.5791132	1.4091408
C9orf84	5.4908	7.33407	11.18177	1.3357015	2.0364555	39.28047	55.00409	52.85768	1.400291	1.3456478
SCARF1	0.81933	1.08894	1.73806	1.3290616	2.1213186	3.53401	5.84977	5.44155	1.6552783	1.5397664
PGAP1	3.50096	4.65298	4.55725	1.3290583	1.3017144	32.99581	56.28404	43.94408	1.7057936	1.3318079
KBTBD3	0.7363	0.97858	1.17144	1.3290507	1.5909819	14.85852	8.49198	10.29494	0.5715226	0.6928644
CTH	87.48736	115.99617	133.83737	1.325862	1.5297909	541.6732	303.07537	301.69501	0.5595169	0.5569686
ZW10	49.77246	65.84974	78.46723	1.3230156	1.576519	337.9155	456.99658	444.83374	1.3523989	1.3164051

NALCN	3.02561	3.99608	5.77644	1.3207518	1.909182	14.05238	19.69095	21.41644	1.4012537	1.5240436
SCN8A	0.54339	0.37916	1.12389	0.6977677	2.0682935	2.81257	1.43249	1.82345	0.5093171	0.6483216
PAEP	4.65547	3.24838	2.96271	0.6977555	0.6363933	13.76937	3.06817	7.81113	0.2228257	0.567283
ABHD16A	4.88976	3.41185	2.07454	0.6977541	0.4242621	13.85971	5.90806	9.40067	0.4262759	0.6782732
SUN3	3.9355	2.74601	0.62613	0.6977538	0.159098	5.81996	19.45256	10.73008	3.3423872	1.843669
HES6	11.96298	8.28098	7.40168	0.6922172	0.6187154	49.14251	27.37556	25.4384	0.5570647	0.5176455
DNAAF1	1.82362	1.24972	2.53868	0.6852963	1.3921102	4.00312	1.50231	5.97606	0.3752848	1.4928506
CXCR4	14.60947	10.0118	4.84238	0.6852952	0.3314549	115.0580	222.6549	160.86212	1.9351532	1.3980957
MUC17	1.02188	0.6945	0.5912	0.6796297	0.5785415	4.59371	2.90816	2.53287	0.6330744	0.5513779
ACSS2	94.96019	64.35581	57.35445	0.6777136	0.6039842	367.5624	200.74633	238.01702	0.5461557	0.6475554
DPEP1	12.97901	8.70785	4.36814	0.6709179	0.3365542	29.0676	9.04726	11.77825	0.311249	0.405202
GCK	0.98578	0.65508	0.26139	0.6645296	0.2651606	4.1001	10.28657	6.20245	2.5088583	1.5127558
SDCBP2	1.987	1.32042	3.16129	0.6645294	1.5909864	9.18267	16.36909	4.86189	1.7826068	0.5294637
C11orf16	1.5579	1.03527	2.47858	0.6645292	1.590975	5.5197	1.71121	3.53965	0.3100187	0.6412758
KCNK7	2.59849	1.72677	0.68902	0.6645282	0.2651617	10.00713	5.70841	3.6332	0.5704343	0.3630611
FGF11	2.18945	1.45495	0.77408	0.6645276	0.35355	4.9467	7.21475	6.76034	1.4584976	1.3666363
SLC26A1	3.78424	2.51473	2.00689	0.6645271	0.5303284	7.72402	3.63707	5.29111	0.4708779	0.6850202

GSG1	1.35066	0.89755	0.35815	0.664527	0.2651667	1.45644	2.22537	3.0688	1.5279517	2.1070556
WNT6	2.38487	1.58481	0.63238	0.6645268	0.2651633	29.39028	6.54892	12.50445	0.222826	0.4254621
SPSB3	1.27238	0.84553	1.68694	0.6645263	1.3258146	6.27214	4.19278	3.78046	0.6684768	0.6027385
HBG2	2.00367	1.33149	0.5313	0.6645256	0.2651634	6.48177	15.40595	3.5019	2.3768122	0.5402691
UNC5C	0.38014	0.25261	0.6048	0.6645183	1.5909928	0.35135	1.04388	0.53151	2.9710545	1.512765
CUX2	0.53105	0.35289	0.70407	0.6645137	1.3258074	0.24542	0.87496	0.37125	3.5651536	1.5127129
SREBF1	276.52162	182.30165	156.87302	0.6592673	0.5673083	724.1736	319.08756	490.148	0.440623	0.6768376
ACAT2	90.06834	59.08168	61.30755	0.655965	0.6806781	510.7865	322.80406	341.44833	0.6319745	0.6684756
TRIM31	58.21195	37.79276	22.8489	0.6492268	0.3925122	270.7886	129.98459	166.46508	0.4800224	0.6147418
FXYD3	7.99888	5.1826	4.13598	0.6479157	0.5170699	40.47751	9.22536	26.42226	0.2279132	0.652764
NOS2	2.26932	1.45417	1.03156	0.6407955	0.4545679	15.13175	22.4338	20.90753	1.4825648	1.3816994
TP53I11	9.9572	6.34114	6.60072	0.6368397	0.6629093	40.64736	18.53231	27.45756	0.455929	0.6755066
IRS4	24.73969	15.58397	13.12014	0.6299178	0.5303276	0.47638	2.26453	1.80161	4.7536211	3.7818758
IDI1	154.65657	97.38434	104.12366	0.6296812	0.6732573	1067.554	645.73771	720.91176	0.6048756	0.6752926
PLCH2	2.51148	1.56464	0.49946	0.6229952	0.1988708	14.15987	7.24141	8.09849	0.5114037	0.5719325
IL34	2.4251	1.51082	0.48229	0.6229929	0.1988743	5.88381	2.99672	3.49673	0.5093162	0.5942969
PHACTR3	0.88847	0.55351	0.53008	0.6229923	0.5966212	2.25828	6.22142	3.02801	2.7549374	1.3408479

SLC16A10	0.97056	0.60465	0.67556	0.6229909	0.6960518	4.42925	2.19878	3.05333	0.4964226	0.689356
F5	0.75653	0.47131	0.15045	0.6229892	0.1988685	0.52443	2.80456	1.38834	5.3478253	2.6473314
STARD4	7.14538	4.39218	3.59992	0.6146881	0.5038109	43.36811	13.73498	22.60386	0.3167069	0.5212092
COL2A1	1.02624	0.61377	0.48982	0.5980765	0.4772958	0.37941	2.70535	0.21523	7.130413	0.5672755
C4B	1.90249	1.13783	0.75671	0.5980741	0.3977472	4.92357	0.31346	2.69331	0.0636652	0.5470238
SOX18	13.35739	7.83209	6.2504	0.5863488	0.4679357	48.65729	28.48075	33.37012	0.5853337	0.6858195
PATL2	1.70255	0.98997	0.90291	0.5814631	0.530328	3.54065	5.6103	6.54638	1.5845396	1.8489204
SV2A	1.68384	0.97909	0.22325	0.5814626	0.1325839	0.25939	0.46239	1.61861	1.7826053	6.2400632
CHRD	11.24767	6.52311	3.90433	0.5799521	0.3471234	53.20823	18.19491	34.74122	0.3419567	0.6529294
SLC1A7	15.65653	9.07714	5.84605	0.579767	0.3733937	58.47431	29.4811	40.54291	0.5041718	0.6933457
LIPE	19.24356	11.14621	9.30897	0.5792177	0.4837447	51.19574	11.13995	23.17943	0.2175953	0.4527609
MMAB	66.41063	38.27426	39.18957	0.5763273	0.5901099	251.5528	112.60234	171.19128	0.447629	0.6805381
NID1	6.00257	3.45192	2.57115	0.5750737	0.4283415	1.06693	2.28229	0.7263	2.1391188	0.6807382
KLHL38	4.78137	2.72345	1.08673	0.5695962	0.2272842	11.3639	20.25736	18.98149	1.7826063	1.6703324
LSS	135.0444	76.4413	72.46091	0.5660457	0.536571	443.3457	264.10481	301.27332	0.5957085	0.6795449
ACHE	18.8735	10.67274	6.35195	0.5654881	0.3365539	78.3315	50.83047	51.57254	0.6489148	0.6583883
SORBS2	1.21382	0.68562	0.57935	0.5648449	0.4772948	2.99173	4.53312	4.49747	1.5152169	1.5033008

ROBO3	1.54149	0.86431	2.60576	0.5606978	1.6904164	1.86999	5.7145	3.94017	3.0558987	2.107054
MMP28	11.06541	6.1513	2.03133	0.5559035	0.1835748	43.8601	16.12792	30.12507	0.3677128	0.6868445
5-Sep	7.0964	3.87909	4.37042	0.5466279	0.6158644	35.65143	15.46381	23.28512	0.4337501	0.6531328
THPO	7.77646	4.2281	4.49899	0.543705	0.5785396	26.46334	4.65913	16.30955	0.1760598	0.6163073
CDHR2	2.78458	1.51399	0.20137	0.543705	0.0723161	9.12497	3.75375	5.97282	0.4113712	0.6545578
RARRES3	12.50634	6.75254	4.97434	0.5399293	0.3977455	44.31054	12.01996	26.22951	0.2712664	0.5919474
MTSS1	2.78168	1.47303	1.2447	0.5295469	0.4474634	4.49931	8.30696	6.38096	1.8462742	1.4182086
AKRIC4	12.73176	6.69799	5.62667	0.5260852	0.4419397	71.58641	16.31549	45.98729	0.2279132	0.6424025
PRKCB	5.12552	2.62176	2.36054	0.511511	0.4605464	24.12323	6.22255	16.90254	0.2579485	0.7006748
MGARP	2.02212	1.00782	0	0.4983977	0	0.9345	9.99504	3.18074	10.695602	3.4036811
CDX2	2.29905	1.14584	0.7838	0.4983972	0.3409234	0.15178	0	0.51662	0	3.4037423
TNFAIP8L2	2.29849	1.14556	0.91421	0.4983968	0.3977437	2.65554	3.78703	1.80773	1.4260866	0.6807391
WNT1	15.99439	7.97155	7.27051	0.4983966	0.4545663	50.15724	24.47029	32.94598	0.4878715	0.6568539
CLRN3	8.22628	4.09995	3.73939	0.4983966	0.4545663	51.05094	19.36251	31.42503	0.3792782	0.6155622
FAM89B	16.71371	8.33005	4.43187	0.4983962	0.2651637	31.32519	16.82865	20.44795	0.5372242	0.6527638
HIST1H3J	6.43141	3.20539	2.55806	0.4983962	0.3977448	15.60399	37.08771	21.91893	2.3768094	1.4047003
SGPP2	2.02909	1.01129	0.64565	0.4983958	0.3181968	4.50105	1.33727	6.80898	0.2971018	1.5127537

CGB5	3.21953	1.6046	0	0.4983957	0	1.48786	0	0	0	0
TFAP2E	2.47384	1.23295	0.98396	0.4983952	0.397746	4.57301	8.15189	6.80976	1.7826093	1.4891199
TNFRSF9	2.37095	1.18167	0.94303	0.4983952	0.3977435	9.58739	25.39167	6.21573	2.6484445	0.6483235
SPDYC	2.73221	1.36172	0	0.4983951	0	0.63133	0	0	0	0
TFF1	2.45701	1.22456	0	0.4983944	0	7.94831	4.0482	5.15307	0.5093158	0.6483227
PDGFA	98.49455	48.60168	40.73243	0.4934454	0.4135501	484.2686	285.87288	323.19684	0.5903187	0.6673916
IFI6	475.79015	230.11171	178.03826	0.4836412	0.3741949	2350.691	1095.9403	1578.8723	0.4662204	0.6716629
MVD	72.03869	34.61643	34.93164	0.4805255	0.4849011	242.4592	102.61431	141.15459	0.423223	0.5821787
SH3BP1	4.31975	2.0453	1.71816	0.4734765	0.3977452	6.78748	3.55865	4.41666	0.5242962	0.6507069
ENO3	75.38695	35.52888	31.86672	0.4712869	0.4227087	261.0743	116.41324	166.87431	0.4459007	0.639183
MSMO1	146.99052	68.59333	68.89169	0.4666514	0.4686812	917.2674	586.94908	554.2219	0.6398887	0.6042097
TNNT2	7.06562	3.1508	1.92285	0.445934	0.2721417	16.41235	5.51437	10.91905	0.3359891	0.6652947
SNED1	15.04013	6.70167	4.83326	0.4455859	0.3213576	71.71541	32.65756	48.56883	0.4553772	0.677244
INHA	40.93096	18.09042	15.3586	0.441974	0.3752319	104.2147	54.71427	68.43257	0.5250145	0.6566494
SQLE	214.78578	93.72616	103.06648	0.4363704	0.4798571	901.9179	540.41229	582.21394	0.5991812	0.6455288
MAGED1	6.52602	2.80392	3.04323	0.4296524	0.4663225	0.51998	0.74154	0.35397	1.4260933	0.6807377
NTN3	4.77174	2.03847	2.16907	0.4271964	0.4545658	17.64157	3.36943	10.7226	0.1909938	0.607803

C7orf55	6.21829	2.65643	2.82663	0.4271962	0.4545671	17.65273	8.78174	10.24698	0.4974721	0.5804757
TCF7	4.17054	1.74333	1.0702	0.4180106	0.2566095	19.0249	7.53643	13.12029	0.3961351	0.6896378
EFR3B	11.69813	4.7433	5.52037	0.4054751	0.4719019	67.34761	14.37385	40.44028	0.2134278	0.6004709
ACPT	5.02528	2.00367	0.79951	0.3987181	0.1590976	8.36052	4.96784	4.74278	0.5942023	0.5672829
GLTPD2	88.666	34.95691	41.05654	0.3942538	0.4630472	220.1685	67.59275	153.34679	0.3070046	0.6964973
AKR7A3	7.12532	2.76207	0.62979	0.3876415	0.0883876	20.1231	10.43537	11.20794	0.5185767	0.5569689
ALDH4A1	4.7734	1.83004	2.04465	0.3833829	0.4283425	22.9081	7.25976	14.63182	0.316908	0.6387182
ACOT1	33.98357	12.99838	12.57379	0.3824901	0.3699961	82.90808	24.7406	48.48267	0.29841	0.5847762
P2RX1	14.32243	5.26871	4.88287	0.3678643	0.3409247	58.15195	24.15973	36.1176	0.4154586	0.6210901
MCAM	30.46987	10.82121	5.95077	0.3551446	0.1953001	147.3049	62.22723	91.26463	0.4224381	0.6195625
C1orf234	6.66902	2.21588	1.76838	0.3322647	0.2651634	5.13666	14.65065	17.48365	2.8521744	3.4037001
HHIPL2	3.70681	1.23164	0.98291	0.3322641	0.2651633	22.46005	12.89341	7.99024	0.5740597	0.3557534
C19orf71	337.44839	106.11549	94.27261	0.3144644	0.279369	1369.179	651.95916	899.40648	0.4761676	0.6568943
PKD2L1	3.5244	1.09784	1.05136	0.311497	0.2983089	9.16175	5.081	12.01154	0.5545884	1.311053
MUC12	76.66062	23.86976	19.04928	0.3113693	0.2484885	229.0381	80.99292	140.43399	0.353622	0.6131467
VTN	29.77333	9.16521	7.31431	0.3078329	0.2456665	110.4794	39.67686	64.73925	0.3591334	0.5859845
CHST13	176.3208	46.01319	60.80012	0.2609629	0.3448267	437.1475	185.77616	294.40935	0.4249736	0.6734782

C11orf71	4.15279	1.03487	0	0.2491987	0	1.91916	6.84221	10.88706	3.5652108	5.6728256
MAPK4	4.63085	1.06523	0.85011	0.230029	0.1835754	23.294	7.92332	15.87582	0.3401442	0.6815412
OR51E2	24.04051	5.16858	7.49961	0.2149946	0.3119572	148.5690	56.69601	100.59306	0.3816139	0.6770796
TOMM20L	6.5507	1.08828	1.73701	0.1661319	0.265164	18.1639	0	3.43469	0	0.1890943
NR4A2	34.46808	5.61286	7.32985	0.1628423	0.2126562	116.3918	55.66562	76.58427	0.4782605	0.6579866

Abbreviations: FPKM, Fragments Per Kilobase of exon model per Million mapped fragments.

Table R2. Relationship between tumor CHIP expression and clinicopathologic features in HCC cohort.

Characteristics	CHIP		P
	Low	High	
Age, years			0.298
≤55	40	115	
> 55	27	57	
Gender			0.767
Female	8	23	
Male	59	149	
Hepatitis history			0.302
No	22	45	
Yes	45	127	
AFP (ng/ml)			0.134
≤ 20	21	72	
> 20	46	100	
ALT (U/L)			0.363
≤ 40	35	101	
> 40	32	71	
GGT (U/L)			0.917
≤ 50	26	68	
> 50	41	104	
Tumor size (cm)			0.042
≤5	30	103	
> 5	37	69	
Tumor number			0.932
Single	55	142	
Multiple	12	30	

Vascular invasion			0.661
No	42	113	
Yes	25	59	
Tumor differentiation			0.269
I-II	43	123	
III-IV	24	49	
TNM stage			0.18
I	38	114	
II-III	29	58	
BCLC stage			0.078
0-A	22	78	
B-C	45	94	

*All the analyses were conducted using Fisher's exact test.

AFP, alpha-fetoprotein; ALT, Alanine aminotransferase; GGT, γ -glutamyl transpeptidase; TNM, tumor-nodes-metastases; BCLC, Barcelona Clinic Liver Cancer;

REVIEWERS' COMMENTS:

Reviewer #2 (Remarks to the Author):

The authors have carefully responded to my comments with additional data and clarification, and improved the quality of the manuscript.

Reviewer #4 (Remarks to the Author):

The Authors have sufficiently addressed the concerns of Reviewer 3

Point-by-point responses to the reviewers' comments

NCOMMS-17-10603B: “The LINC01138 Drives Malignancies via Activating Arginine Methyltransferase 5 in Hepatocellular Carcinoma”.

REVIEWERS' COMMENTS:

1. Reviewer #2 (Remarks to the Author):

The authors have carefully responded to my comments with additional data and clarification, and improved the quality of the manuscript.

Response: We greatly appreciate the reviewer's nice comments.

2. Reviewer #4 (Remarks to the Author):

The Authors have sufficiently addressed the concerns of Reviewer 3

Response: We greatly appreciate the reviewer's nice comments.